# Arithmetic optimization based secure intelligent clustering algorithm for Vehicular Adhoc Network

**Asad Ali**[1]*, **Muhammad Assam**[2], **Masoud Alajmi**[3], **Yazeed Yasin Ghadi**[4],
**Salgozha Indira**[5], **Ainur Akhmediyarova**[6], **Tahani Jaser Alahmadi**[7],
**Hend Khalid Alkahtani**[7]*

**1** Department of CS and IT, University of Engineering and Technology, Peshawar, KPK, Pakistan,
**2** Department of SE, University of Science and Technology Bannu, Bannu, KPK, Pakistan, **3** Department of CE, College of Computers and Information Technology, Taif, Saudi Arabia, **4** Department of CS and SE, Al Ain University, Abu Dhabi, United Arab Emirates, **5** Department of Informatics, Abai Kazakh National Pedagogical University, Almati, Republic of Kazakhstan, **6** Department of Information System, Satbayev University, Almaty, Kazakhstan, **7** Department of Information System, College of Computer and Information Sciences, Princess Nourah bint Abdulrahman University, Riyadh, Saudi Arabia

* Hkalqahtani@pnu.edu.sa (HKA); asad.ali@uetpeshawar.edu.pk (AA)

**Data Availability Statement:** All relevant data are within the article and its Supporting information files.

**Funding:** This work was supported by the Princess Nourah bint Abdulrahman University Researchers

## Abstract

Vehicular Adhoc Network (VANET) suffers from the loss of perilous data packets and disruption of links due to the fast movement of vehicles and dynamic network topology. Moreover, the reliability of the vehicular network is also threatened by malicious vehicles and messages. The malicious vehicle can promulgate fake messages to the node to misguide it, which may result in the loss of precious lives. In this situation, maintaining efficient, reliable, and secure communication among automobiles is of extreme importance, especially for a densely populated network. One of the remedies is vehicular clustering, which can effectively perform in a high-density network. However, secure cluster formation and cluster optimization are important factors to consider during the clustering process because non-optimal clusters may incur high end-to-end communication delays and produce overhead on the network. In addition, malicious nodes and packets reduce passenger and driver safety, increase road accidents, and waste passenger and driver time. To this end, we employ Arithmetic Optimization Algorithm (AOA) to design a secure intelligent clustering named AOACNET. AOA is used to achieve optimality of vehicular clusters. During cluster formation, the algorithm prevents unauthentic nodes from becoming cluster members by taking into consideration the performance value of each automobile. The vehicle's performance value is based on the record of data transmission. If a vehicle transmits a fake message, it will receive a penalty of (-1), and in the case of transmitting a legitimate message, a reward of (+1) will be assigned to the vehicle. Initially, all the vehicles have equal performance value which either increase or decrease based on communication with their peers. The vehicles will become cluster members only if their performance value is greater than the threshold value (0). AOACNET is tested in MATLAB using various evaluation metrics (i.e., number of clusters, load balancing, computational time, network overhead and delay). The

Supporting Project number (PNURSP2023R384), Princess Nourah bint Abdulrahman University, Riyadh, Saudi Arabia The funders had no role in study design, data collection and analysis, decision to publish, or preparation of the manuscript.

**Competing interests:** The authors have declared that no competing interests exist.

simulation results show that the proposed algorithm performs up to 25% better than the similar contenders in terms of designated optimization objectives.

## Introduction

The analysis of a problem within a given constraint for the best solution is called optimization. The optimization process involves the maximization or minimization of an objective function to get the optimal decision variable. There are numerous real-world optimization problems not only in the field of computer science but also in other fields like electronics, biomedical, engineering, accounting etc. [1]. Due to the ever-increasing complexity of real-world optimization problems, it is very challenging for deterministic optimization techniques such as fast steeper [2] and the quasi-Newton method [3] to tackle complex optimization problems. Therefore, there is always a need for efficient optimization techniques, such as meta-heuristic algorithms [4]. Meta-heuristic algorithms leverage stochastic operators, which introduce randomness, to aid the search agents in estimating optimal solutions. These algorithms employ gradient-free methods and avoid local optimal stagnation to find the optimal decision variable. Such characteristics make them an efficient tool to tackle complex, non-convex, non-linear, and non-continuous optimization problems.

This paper examines clustering as a combinatorial optimization problem using a class of meta-heuristic algorithms for Vehicular Adhoc Networks (VANET). VANET have potential applications that contribute to the development of intelligent transportation systems (ITS). Under the umbrella of a vehicular network, automobiles communicate with each other to escalate road safety and transportation efficiency [5, 6]. The communication between the vehicles may include hazard warnings, security threats, location, speed, direction, traffic congestion information, and weather information. Automobiles communicate with each other and with the Roadside Units (RSUs). The former is called vehicle-to-vehicle, and the latter is called vehicle-to-infrastructure. VANET applications are classified as comfort applications, safety applications, and non-safety applications [7]. Automobiles use Wireless Access for Vehicular Environments (WAVE) as a communication standard [8]. The WAVE standard uses a total of 75 MHz of capacity, which is divided into seven channels each of 10 MHz capacity, i.e., six service channels and one control channel. The service channel is reserved for comfort applications, and the control channel is dedicated to safety-related beacons. The spectrum allocation in the WAVE standard is shown in Fig 1.

VANET exhibits a dynamic network topology due to rapid movement of automobiles, resulting in disruptions of connections between vehicles and the loss of critical safety data packets. Consequently, the complexity of designing an efficient communication routing protocol becomes more intricate, especially in a densely populated network. The dynamic environment of vehicular networks is a critical challenge for safety-related applications to ensure efficient, timely, and reliable delivery of data packets [6]. Along with the communication between cars, the vehicular network have to tackle various tasks, like reducing the communication latency to enable the vehicles to take the right decision at the right time, overcoming frequent network disconnection to get intermittent connectivity, network security, i.e., protecting from malicious nodes and messages, reliability, investigating the source of the message to confirm it is from a reliable node, resource management, and scalability, which allow the network to use the scarce wireless resources efficiently even for a higher density of vehicles [9].

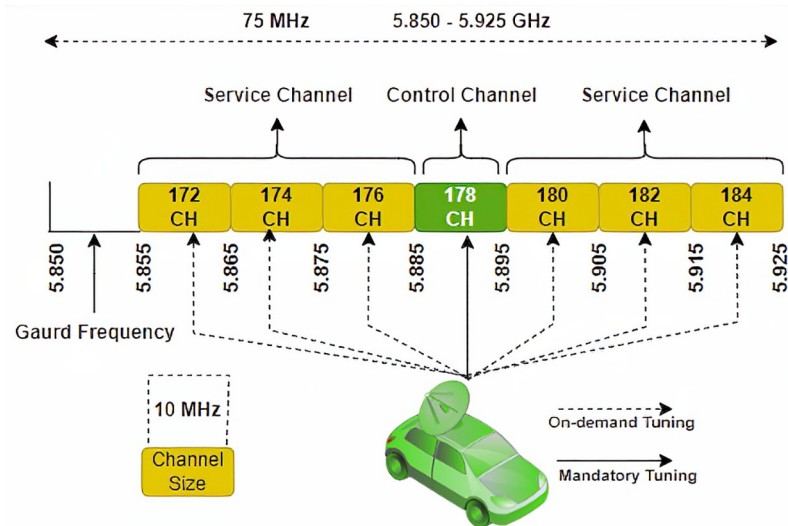

**Fig 1. Spectrum allocation in WAVE standard.**

Regarding the security of vehicular network, the identification of authentic node is crucial to protecting the vehicular network from malicious vehicles and messages [10]. The malicious vehicle can propagate the fake messages to nodes to misguide them. Fake messages can take variety of forms like reporting fake events on the road, indicating fake route as best route, false reporting vis-à-vis weather information, fake neighbor identification, etc. Consequently, malicious nodes and packets reduce passenger and driver safety, increase road accidents, and waste passenger and driver time [11]. In most cases, malicious messages are generated by the fake nodes but this is not the case all the time, in some cases, fake messages can also be generated by the authentic node, which is one of the most challenging task to detect. Therefore, it is necessary to escalate transportation efficiency by crafting an efficient dynamic communication protocol with the capability to attain high levels of network security, reliability, and scalability.

To this end, one of the remedies is secure vehicular clustering, where vehicles are grouped into logical sets based on certain criteria (e.g., speed, distance, position, etc.). Each logical set is lead by a cluster head (CH), while to perform ground-level operations, there are set members called cluster members. Clustering helps to escalate communication efficiency by reducing the direct communication links between automobiles, making the vehicular network more manageable [12, 13]. Vehicular clustering supports localized communication where communication occurs with the nearby vehicles. This increase the stability communication links and reduces the effect of dynamic network topology. Vehicular clustering enables efficient allocation of scarce wireless resources, e.g., bandwidth, to the members of the clusters based on their communication needs, resulting in optimizing resource utilization. Clustering increases communication reliability and signal quality by minimizing interference between communication channels. Moreover, clustering can be employed to obtain optimal routing paths, making vehicular networks more efficient [14, 15]. Vehicular clustering provides an efficient environment for the cluster members to collaborate for specific tasks such as group-based navigation, collaborative collision warning, cooperative sensing, etc. Vehicular clusters are independent in terms of functionality and operation that increases network resilience, i.e., if one cluster experiences communication issues, the other clusters operate independently without being effected. Vehicular communication via clustering is shown in Fig 2.

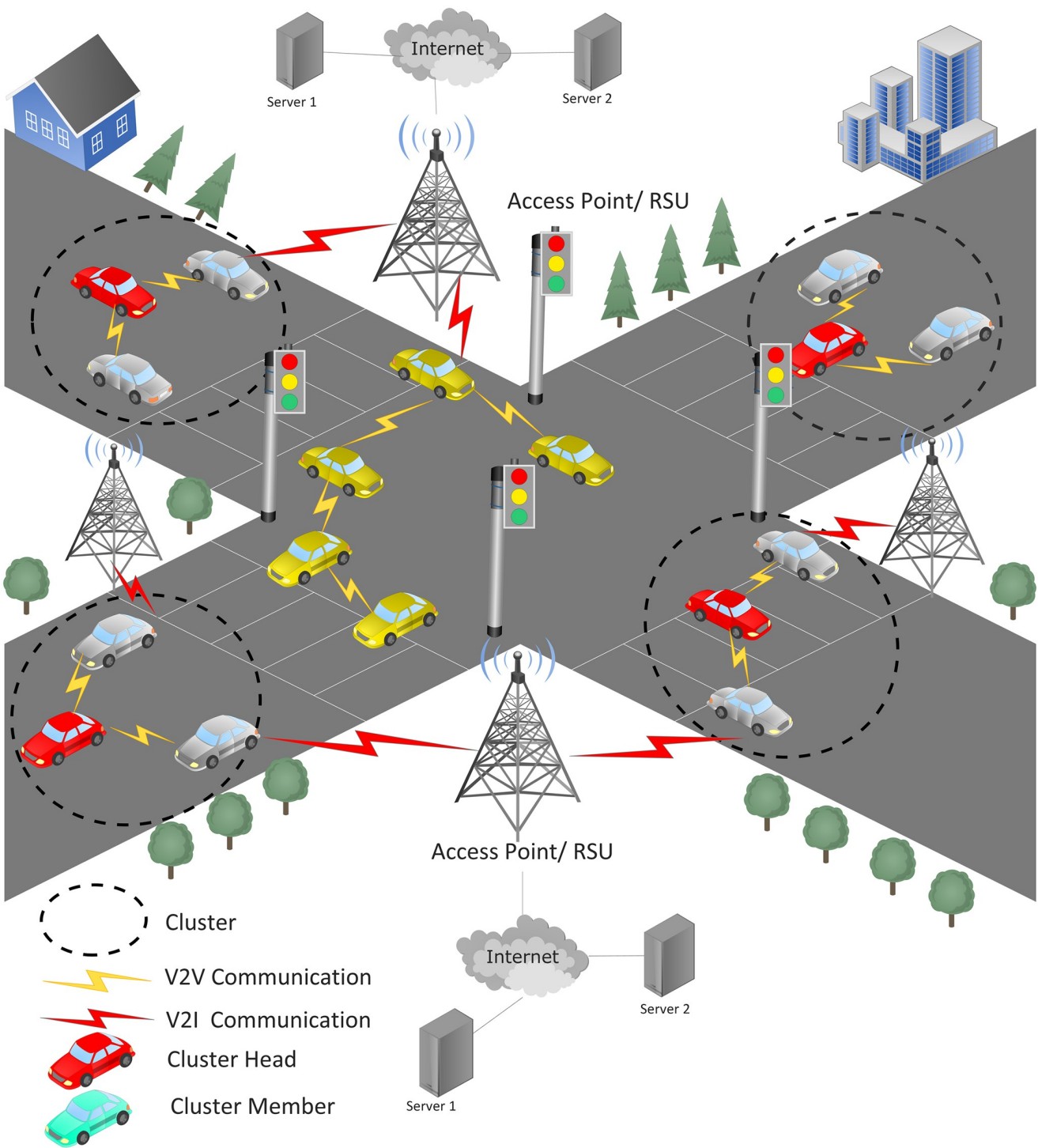

**Fig 2. Vehicular communication via clustering.**

Various challenges affect the efficiency of clustering schemes, e.g., cluster stability, vehicle's mobility patterns, communication overhead, number and size of clusters, and cluster security. Within the cluster, each member of a cluster may have a different requirement for QoS. It is desirable to provide an adequate resources to diverse application. clustering schemes must be able to allocate channel efficient to avoid channel congestion and interference. Numerous clustering-based communication protocols are proposed to address different issues. However, one of the challenging tasks in clustering is finding the optimal number of clusters i.e., too many CHs for a given vehicular network, which incurs high end-to-end delay. The large number of CHs also produces significant communication overhead on the control channel, resulting in network congestion and loss of safety data packets [16]. In addition, during cluster formation, preventing unauthentic vehicles as a cluster member and stopping fake messages from authentic or fake vehicles become intricate towards the escalation of transportation efficiency. Hence it is crucial to obtain optimality of CHs and cluster security during cluster formation for efficient and authentic data dissemination between automobiles. A network with optimal clusters have high throughput, security, and reliability, more manageable, scalable, and low latency. The communication overhead on the control channel is also reduced because of the creation and maintenance of optimal clusters. Optimal clustering significantly improves the traffic management by coordinating with the nearby vehicles in the congested area. Optimal clusters efficiently allocate communication channel to the members that contribute to the efficient channel access mechanism. As a result, overall communication efficiency is improved. Optimal number of vehicular clusters are adaptive to vehicular network dynamics i.e., clusters are formed based on the vehicular density, network condition and mobility patterns [17, 18].

Moreover, due to secure clusters, the risk of different malicious attacks (e.g., masquerade attacks, DOS attacks, reply attacks, release of message content attacks, etc.) is minimized. The trust level of automobile communication have also increased, which benefits society and the community. By averting unauthentic nodes and fake messages, the vehicular network protects user privacy and escalates transmission reliability and accuracy. Secure vehicular clustering helps the network to protect against Sybil attacks that limit the influence of individual nodes in the cluster. Moreover, secure clustering improves the utilization of scarce wireless resources by optimizing communication and data dissemination. Secure vehicular clustering improves traffic control systems and minimizes false collision warnings. A malicious node can propagate the manipulated information to the CH, which influences the traffic control system. For instance, a rough node could wrongly claim the emergency to prioritize its movement. This results in the disruption of traffic flow and potential collision at the intersection point [19–21]. Recently, various vehicular clustering algorithms have been proposed to optimize vehicular clusters, such as GWOCNET which imitates the group hunting maneuver of wolves to obtain optimality of vehicular clusters [22], CACONET [23] modeled the food searching behavior of ants using pheromone and heuristic values to optimize clusters, and CLPSO [24] is a variant of particle swarm optimization that enhances the searching capability of particles within the search space using local best search and global best search strategies. The algorithm is able to obtain optimal vehicular clusters with the cooperation of both local and global best search strategies. All of these clustering algorithms optimize the vehicular clusters to attain elevated levels of Quality of Service (QoS) requirements. However, due to minimum stochastic variables, a low convergence rate, and various search spaces, the algorithms fall into local stagnation, resulting in a decrease in their efficiency in finding the best optimal solution. In addition, none of the existing clustering algorithms consider the security of cluster formation, which also reduces the reliability and privacy of message transmission. In this paper, we exhibit the use of AOA to achieve the optimality of CHs for efficient vehicular communication. AOA is a new nature-inspired meta-heuristic algorithm that takes inspiration from the distribution

behavior of arithmetic operators (i.e., division, multiplication, addition, and subtraction) in solving arithmetic problems [25]. These operators are the basic element of number theory and are used as fundamental calculation measures to examine different arithmetic problems.

AOA is considered the best optimizer to tackle various optimization problems particularly vehicular clustering because of its straightforward implementation. It does not require setting different parameters except swarm size, control parameters, and termination criteria. The stochastic and adaptive variables of AOA escalate the process of convergence and divergence of the search agent solution. The distribution behavior of AOA increases the capability of global searching which leads to obtaining global solutions. Because of minimal parameter tuning, AOA is a flexible and adaptive algorithm that enable it to adapt to dynamic environment of vehicular network. AOA can dynamically adjust the search process to obtain the global optimal solution. Moreover, the exploratory and exploitative behavior of AOA is balanced that help the agent to obtain potential optimal solutions. Such prominent characteristics of AOA make it effective method to solve vehicular clustering problem.

In the proposed solution, the optimization of secure vehicular clusters using AOA encompasses three phases i.e., solution construction i.e., formation of secure clusters, solution evaluation using the objective function, exploration, and exploitation phases. The exploration behavior randomly searches the potential areas for an optimal solution by using Division (D) and Multiplication (M) search strategy while the exploitation phase inspects the potential areas found in the exploration phase by using Addition (A) and Subtraction (S) search strategy. In the first phase, secure clusters are formed by employing various attributes of the vehicle like vehicle number, location, vehicle model, vehicle manufacture, and vehicle performance record, which is a record of receiving a reward or penalty for each message transmission. If a vehicle transmits the wrong message, a penalty of (-1) is assigned to this transmission, and in another case, a reward of (+1) will be assigned. If a vehicle performance record meets the threshold by receiving more penalties, then the algorithm declares the vehicle as unauthentic. After secure cluster formation, the solution evaluation phase is performed using the fitness function, which gives us the optimal solution in the current iteration. Afterward, the position of each candidate solution is updated to get closer to the more optimal solution. After the last cycle, the best candidate solution have the optimal secure clusters.

The proposed algorithm is simulated in MATLAB and compared with two different category of algorithms i.e., clustering algorithms (e.g., CLPSO, GWOCNET, and CACONET) and authentication schemes (Sun's authentication scheme [26], Khan's authentication scheme [27], Yanwei's authentication scheme [28] and Subramani's authentication scheme [29]). Due to the infusion of stochastic operators and an equilibrium phase of exploration and exploitation behaviors within the AOA, substantial performance escalation is witnessed in our proposed algorithm. This enhancement is in the form of a significant reduction of end-to-end delay, maximizing message reliability, accuracy and security.

## Contribution

Although various clustering schemes using evolutionary algorithms are proposed to optimize the network performance. However, due to local stagnation and premature convergence, the algorithms may not be able to get the desired output and create a space for further improvements. So our contributions to this paper are as follows:

- We use performance attribute to secure the clusters by preventing malicious and unauthenticated nodes from becoming cluster heads and cluster members.

- We model the distribution behavior of arithmetic operators to optimize secure clusters.

- We designed a multi-objective fitness function to cope with the complexities of the clustering process in terms of evaluation.

- We performed an extensive comparison using different performance metrics like load balancing and a number of secure clusters to demonstrate the efficacy of the proposed algorithm as compared to similar approaches.

- A numerical performance analysis is also made to show the results in percentage.

TThe rest of the paper is organized as follows: The literature review section discusses the extensive background of the various clustering protocols. This section also presents the details of the Arithmetic Optimization Algorithm (AOA). The proposed Solution section discusses the proposed algorithm along with the flowchart. The experimental setup and result discussion section presents details regarding the simulation environment, contains results discussion and analyze the security of the proposed algorithm against well known security attack. Finally Conclusion section concludes the article with future direction.

## Literature review

In literature, numerous applications are designed where clustering protocols can work efficiently and escalate the performance of the vehicular network. Broadly we can categorize vehicular clustering protocols into three dimensions i.e., general clustering protocol, domain-specific clustering protocol and meta-heuristic based clustering algorithm. General clustering protocols focus on basic parameters of clustering i.e., forming clusters with robustness and reliability. Such protocols are not designed for specific domains like topology management, routing, resource optimization security, etc. [17]. Domain-specific clustering protocols focus on substantial performance enhancement of specific applications like routing, channel access mechanism, etc. Domain-specific clustering protocols are further categorized into optimization, security, routing, channel access management, topology management, and Quality of Service (QoS) assurance [30]. Meta-heuristic based clustering algorithm uses social behavior and feeding maneuver of animals as inspiration to optimize various parameters of clustering schemes. Particle Swarm Optimization, whale optimization algorithm and ant colony optimization are few examples of meta-heuristic algorithms. Fig 3 shows the taxonomy of clustering protocols for vehicular network.

### General purpose clustering protocol

The primary objective of the general-purpose clustering algorithm is to create stable and robust vehicular clusters by taking into consideration mobility patterns, vehicle speed, and vehicle density as system parameters. To this end, researchers are finding advanced methods for cluster formation, efficient techniques for CH selection, and an effective way to join or leave the proximity of clusters. One of the first general clustering algorithms is density-based clustering, where nodes are portioned on the basis of the vehicle's density. The objective is to minimize cluster head changes. To this end, vehicles join the part of the network by determining the connectivity level. The next step measures the link quality and relative speed of vehicles to select stable links. Finally, the node's performance record is examined to develop its reputation. The mobility-aware clustering scheme is proposed by Morales et al. [31]. The algorithm uses destination location to accurately follow the mobility pattern. The algorithm takes into account the vehicle speed, location, direction, and relative destination as system metrics for cluster creation and cluster head selection. One of the complex tasks in this algorithm is determining the relative position of the destination due to random and dynamic vehicular network

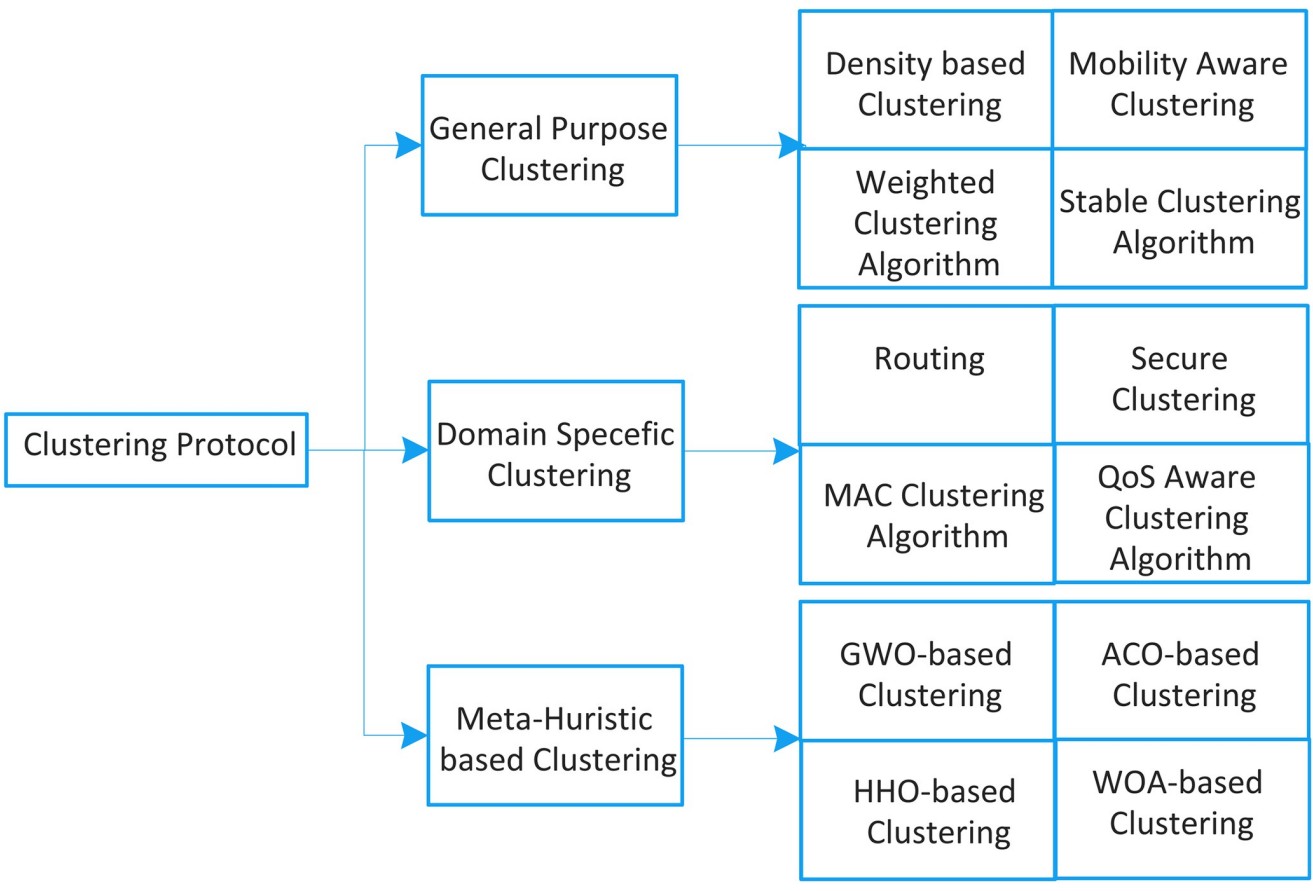

**Fig 3. Classification of clustering protocol.**

topology. The algorithm considers the same average speed of each vehicle that may not be the situation in real scenario. This may result in the poor performance for some applications.

An intelligent learning automata-based clustering algorithm is proposed in [32] that uses link counts and node mobility to form the clusters. The algorithm first selects the leaders and then CH among the potential leaders. The leader is selected based on the aggregate local connectivity of each vehicle. If the aggregate local connectivity of the vehicle is greater than the threshold value, then the vehicle is selected as the leader. The CH is selected among the leaders that have the maximum aggregate degree of connectivity. The paper employed an innovative approach to perform clustering using the combination of learning automata and ant swarm intelligence. This increases the local and global search ability of each ant, that enhances the quality of clustering results.

A weighted clustering algorithm is proposed in [33] where CH is elected based on various simulation parameters such as location, battery power, transmission range node mobility, etc. A weight is assigned to each vehicle based on system parameters, and the node with the highest weight is elected as CH. The weighted clustering algorithm minimizes communication and computation costs by dynamically electing a CH. Moreover, the algorithm optimizes the MAC operation by keeping the number of members in a cluster within the threshold value. The algorithm performs dynamically i.e., where there is a need and distributes the load across all clusters.

Another stable clustering algorithm for VANET is proposed by Cheng et al. [34]. The clustering algorithm determines the center regions of high-density vehicles with the help of the beaconing process. The clustering algorithm handles the crucial issue of management overhead in vehicular networks with an emphasis on stable and quality communication. A blob detection method [35] is used to form clusters based on the detected center regions. All the vehicles are equipped with LTE and 802.11p communication interface.

## Domain-specific clustering protocols

Domain-specific clustering protocols focus on substantial performance enhancement of specific domain applications. The objective is to determine the areas where clustering schemes can escalate the performance of the network [36]. There are various domains of vehicular networks where clustering intends to attain elevated levels of efficiency, like routing, channel access mechanisms, security, resource management, and optimization.

**Cluster-based routing protocol.** Routing in a vehicular network means finding a reliable path for data delivery from source to destination. Clustering can be effectively employed to offer an optimal path. Various performance metrics are used to examine the function of vehicular routing protocols, like throughput, end-to-end delay, packet loss, and delivery ratio. Clustering plays an important role in escalating routing efficiency in a vehicular network by grouping vehicles into clusters. This results in more localized and stable communication and reduces contention for shared scarce resources. Clustering reduces the routing overhead associated with maintenance of routing updates. Instead of broadcasting information, nodes within a cluster can only communicate with the cluster head. Efficient clustering protocols optimize different routing decisions by mitigating the effects of high mobility and a dynamic environment. The cluster-based routing protocols claim to achieve high performance in terms of reliability and scalability as compared to traditional routing protocols such as [36, 37].

An intelligent clustering scheme is designed by Mehrnaz et al. [38]. The primary objective of the algorithm is to address the scalability issue and increase the packet delivery ratio and throughput. The algorithm also uses the RSU as CH if it is within the transmission range of the clusters. In other case, CH is elected based on the training of an artificial neural network using a genetic algorithm. Cluster size, vehicle density, and bias are taken into consideration as system parameters. An enhanced clustering-based routing protocol is proposed by Afia et al. [36]. The protocol primarily focuses on improving network stability and average throughput using the Sugeno model fuzzy inference system. The fuzzy inference system is used to help with the selection of optimal CH. The inputs for the fuzzy inference system are vehicle distance from the RSU, distance between vehicles, the vehicle's neighbors (node degree), residual energy, and concentration.

A path pattern discovery-based clustering protocol for VANET is proposed in [37]. The protocol addresses the dynamic random topology of vehicular net-works that causes frequent connection interruptions and reduces communication efficiency. To this end, the authors use the vehicle's path prediction as the primary method to send data from source to destination. The history of a vehicle's movement is stored on the road to predict the movement pattern for new cars. More data is needed to increase the accuracy of the prediction. This can increase the accuracy of prediction on congested roads but reduce the efficacy of the proposed algorithm on empty roads.

**Secure clustering protocol.** Vehicular networks are susceptible to various threats and attacks due to their ad hoc architecture and spontaneous nature. The attacks can raise serious issues such as wasted time and resources and, sometimes, lifesaving issues. Therefore, an effective approach to safeguarding vehicular networks from cyberattacks is necessary. A Sybil attack

detection and avoidance scheme is proposed for a clustered vehicular network to protect the net-work from fake messages generated by Sybil nodes [39]. To detect the Sybil attacker, CH calculates the signal strength of each received packet from each node and then compares the received signal strength index value for a period of 10 seconds using the Longest Common Subsequence (LCSS). CH declares nodes as Sybil attackers if they have the same pattern of received signal strength index values. In the other case, if nodes use a power control mechanism, then the authors uses a point detection mechanism to calculate the mean and standard deviation within each segment of the time series. The algorithm enhance vehicular security using the natural dispersion of platoons over time and analyzing the time series data of vehicle movement. A vehicle that travel closely in cluster for unusual extended duration but follows unexpected of pattern of data streams is considered as potential Sybil node.

A CNN-based trust-aware clustering protocol is proposed by Annamalai et al. [40]. Data authentication within the clustering approach is also introduced using the quantum cryptography technique. The trust degree of nodes is calculated using a convolutional neural network by taking into account the distance between nodes, node direction, and speed of vehicle as system parameters. The proposed protocol address various factors to enhance vehicular security like quantum cryptography is used for data authentication that ensures high level security. Data transmission is prioritise to protect critical data streams from eavesdropping.

In [41], the authors proposed Blockchain and quantum mayfly optimization-based secure clustering schemes for vehicular networks. The formation of clusters and the selection of optimal CHs is performed using the Mayfly optimization algorithm. To safeguard cluster head and cluster member privacy and to provide authentic and trustworthy services to the users, Blockchain technology is used. Moreover, Blockchain technology also ensures the integrity of data transmission between cluster head and cluster members. The protocol employs Blockchain technology as a trust infrastructure that leverages the immutable and decentralized structure of blockchain to build trust, that ensure the integrity of cluster formation process.

A cluster-based intrusion detection approach for VANET is proposed by Muthumeenakshi et al. [42]. This is an adaptive approach where the process of cluster formation is done by sparse fuzzy C-means clustering algorithm, and intrusion detection is performed by an adaptive elephant fuzzy system. Each CH uses an adaptive fuzzy system that is responsible for detecting intruders within the vehicular network to ensure the security of vehicular communication. An authentication-enabled cluster-based routing protocol is proposed by Azhdri et al. [43]. Date communication between source and destination is divided into three phases: clustering, routing, and authentication. The authentication between CH and cluster members is performed using a message authentication code and symmetric key cryptography. The protocol uses end-to-end delay, packet delivery ratio, packet drop rate, and throughput as evaluation metric and outer perform benchmark algorithms. However, routing overhead have slighty increased.

P Bagga et al. [44], proposed a Blockchain-based batch authentication scheme for the Internet of Vehicles (IoV). The authors proposed two authentication schemes i.e., V2V authentication and batch authentication. In V2V authentication, each vehicle authenticates its neighbor vehicle within a cluster while in batch authentication, the cluster vehicles are authenticated as a group by the RSU. The data is securely gathered by the RSU and formed into blocks by fog and cloud servers. In [45], Blockchain-based mutual-healing group key distribution is proposed to distribute the group key among the nodes securely. A private Blockchain is designed by the ground control station where the group keys messages are stored and distributed. The Blockchain also recorded the network member's joining and revoking as a transaction. The authors in [46], proposed a Privacy-Preserving Reputation Updating scheme for the cloud-assisted vehicular networks to resolve the bottlenecks faced by the trusted authorities during

reputation management. The authors employ Elliptic Curve Cryptography (ECC) and Paillier cryptosystem to preserve privacy and security during preprocessing reputation feedback. In [47], the authors proposed a context-aware trust management model for a vehicular network to evaluate the trustworthiness of received messages based on context-specific information without effecting the trust level of network entities.

**Cluster-based medium access control protocol.**   The fast mobility of vehicles, scarce wireless spectrum and random distribution of automobiles make vehicular network more challenging environment to perform fair channel allocation process. To this end, various cluster-based MAC routing protocols are proposed to address the issue of fair channel access. The focus of these protocols is to address the fast mobility of vehicles and the dynamic network topology. clustering reduces the complexity of fair channel access because of the coordination between cluster head and cluster members. Within a cluster, the cluster head is responsible to allocate channel based on some strategy like fixed and dynamic time slot assignment, reservation strategy etc. Cluster-based TDMA protocol is proposed by Almalag et al. [48] with an objective to improve network performance in terms of high throughput, low delay, high packet delivery ratio and low packet drop. Within a cluster, a time slot is assigned to each vehicle. Low priority vehicles utilize its slot by transmitting packets for high priority vehicles while the low priority transmission is manage by RSUs.

Contention-free cluster-based MAC protocol is proposed in [49] with an objective to tackle collision issues within a cluster and intra-cluster transmission. CH uses TDMA to assign time slot to cluster members based on their needs. Interference within a cluster is reduced by minimizing access delay and comparison of relative offset time during slot assignment.

Cluster-based MAC protocol is proposed by et al. [50]. The authors modify the IEEE 802.11p standard for vehicular communication as primarily it is design for direct communication not for cluster-based communication. Therefore new control packets are add-ed and existing control packets modified to be able to perform cluster-based communication. To eliminate the hidden node problem, request to send (RTS) and clear to send (CTS) packets are added to non-safety data dissemination.

To minimize the problem of cluster rupture due to turbulent traffic situation, motion parameters-based cluster medium access protocol is proposed in [51]. Vehicles organized themselves into clusters. The vehicles in one cluster may use the idle channel of another cluster. The hidden station problem is avoided by RSU's that periodically monitors the utilization of channels and advertise them. The protocol escalate the channel utilization by allowing the use of ideal channel in high-traffic situation.

**Quality of Service (QoS) aware clustering protocol.**   Frequent disconnection of links in vehicular networks is considered a challenging task due to the rapid mobility of vehicles. Various studies have been proposed to ensure the QoS by addressing issues caused by frequent link disconnections such as delay, network congestion, throughput, packet drop, etc. One of the remedies is QoS-aware clustering, which plays a significant role in maintaining the QoS metrics that contribute to substantial network performance improvement. Clustering improves the reliability and responsiveness of the vehicular network by optimizing the utilization of resources. Clustering provides an efficient channel allocation process that reduces channel contention and packet collisions that positively impact the QoS metrics related to reliability and packet drop. Clustering divides the network into different partition, if one partition experiences communication issues, the other partitions can operate normally. This positively impact the QoS metric related to fault isolation and availability. A QoS-aware CH selection algorithm is proposed for VANET in [52] to improve network throughput, reduce energy consumption, minimize packet loss, and achieve a high packet delivery ratio. Clusters are formed based on QoS metrics such as the vehicle's density, speed, velocity, and distance.

QMM-VANET is another clustering protocol proposed by Hamidah et al. [53] that takes QoS metrics as simulation parameters. The objective of the protocol is to ensure efficient communication by forming reliable and stable clusters. The protocol divides the operation into three phases i.e., CH selection by considering QoS parameters, selecting nodes as gateway, and selecting another gateway in case of link failure. The performance analysis reveals efficient results of the QMMVANET protocol in terms of low communication latency, maintaining network stability, and high packet delivery ratio. Maintaining network topology in VANET incurs high network overhead on the control channel that affects the communication process. To reduce overhead on the network, in [54] the author proposed a QoS-aware cluster-based routing protocol for VANET. To ensure efficient communication via clustering, new control packets are introduced and existing control packets of IEEE 802.11 are modified. The protocol uses a hybrid scheme i.e., it transmits time-sensitive data via QoS shortest path and normal data via CH. In this way, topological overhead is reduced. The proposed algorithm is based on the IEEE 802.11 EDCA mechanism to ensure QoS at the MAC layer. The results indicate the efficient performance of the algorithm in terms desired throughput and QoS.

## Meta-heuristic based clustering algorithms

Meta-heuristic algorithms are effective tools to tackle complex optimization problems for multiple reasons, like problem independence, use of stochastic operators, self-adaptability, and a high convergence rate to prevent local optima. Meta-heuristic algorithms are inspired by the social behavior and feeding maneuvers of animals [55], nature phenomena [56], and human behavior [57]. The two important characteristics of the meta-heuristic algorithms are diversification (exploration) and intensification (exploitation), which enable them to examine feature space for an optimal solution. Diversification employs the search space extensively for diverse solutions, while intensification goes for an optimal solution by in-depth analysis of the diverse solutions found in the diversification phase.

Clustering is deliberate as a combinatorial optimization problem. Numerous clustering algorithms are proposed by employing meta-heuristic algorithms to increase the performance of the network. In [58], the author proposed a genetic algorithm-based clustering algorithm to minimize network overhead on the control channel. Crossover and mutation methods play an important role in generating a new solution from diverse solutions and avoiding premature convergence. The proposed algorithm aims to achieve an equilibrium state between the operation of Medium Access Control (MAC) and minimum number of clusters, resulting in more scalable and efficient mobile network. Moreover, the algorithm evenly distribute the load on each cluster, which increase the lifetime of the network. In [59], the same author employs simulated annealing to optimize the performance of the algorithm by obtaining an optimal number of clusters for a given network in an attempt to reduce communication latency and network overhead. By employing simulated annealing for clustering, the reliability and scalability of the network is improved. In [24], the authors use a variant of the Particle Swarm Optimization algorithm i.e., comprehensive learning particle swarm optimization to perform optimization of clusters by taking into contemplation the battery power, ideal degree, mobility, and transmission range as system parameters. The optimization process starts with the initialization of particles encoded as candidate solutions within the problem space. The objective function is used to examine the candidate solutions and then the velocity of particles is updated to incorporate diversity in the solution. The algorithm attempts to minimize the number of clusters that result in efficient utilization of resources and improve routing costs and decisions. Consequentially, this improves reliability and reduces latency and hence overall network performance is improved. To generate multiple solutions at each run via Pareto optimal

front instead of just a single solution, a multi-objective particle swarm optimization-based clustering algorithm is proposed in [60] with an objective to effectively utilize the wireless spectrum by forming optimal clusters and minimizing the energy consumption. Degree difference, ideal degree, transmission range, and grid size are taken as system parameters.

In [23], the authors employ an ant colony optimization (ACO) algorithm for intelligent clustering in Vehicular named CACONET. ACO is inspired by the food-searching maneuver of ants. While searching for the food, ants release a chemical fluid called pheromone. The intensity of the pheromone quantity on the path directs the ants to follow that specific path. Intelligent clustering starts with the initialization of search agents (candidate solution) and vehicles along with the pheromone and heuristic value within the search space. The fitness value of each candidate solution is calculated using the fitness function. The fitness value is strongly related to the pheromone and heuristic values. The path that have the best fitness value contains optimal CHs. The algorithm optimizes the vehicular clusters, resulting in more stable communication and reducing packet routing costs and overhead. This improves the scalability and reliability of the vehicular network.

In [61], the authors compute the load on each cluster along with the computational complexity of the proposed algorithm.

In [22], the authors exhibit the use of the Gray Wolf Optimization (GWO) algorithm for intelligent clustering with the objective of attaining an elevated level of scalability and reliability. GWO emulates the feeding strategy of wolves to form the optimized clusters. GWO is a stable and adaptive algorithm that facilitates the creation of stable and optimal clusters by the proposed solution as it adjusts to the dynamic network. By using GWO for clustering, the vehicular network's communication efficiency is increased, ensuring reliable packet delivery. Ali et al. [62] designed a clustering scheme using a whale optimization algorithm. The proposed algorithm mimics the feeding maneuver of whales to form intelligent clusters with an objective to minimize latency and network overhead. The same author also exhibits the use of the Harris Hawks Optimization (HHO) algorithm to form optimized and balanced clusters in an attempt to minimize communication latency and overhead. Moreover, the numerical analysis and the computational complexity of the algorithm are also provided [63]. HHO-based clustering improves the scalability of the network by efficiently utilizing the scarce wireless resources and maintaining and optimized clusters. Although various clustering algorithms using bio-inspired methods are proposed but still there is space for improvement in terms of attaining a high level of reliability and scalability. Moreover, none of the bio-inspired clustering algorithms consider security of the cluster formation process which is a main factor in achieving high data reliability and accuracy.

## Arithmetic Optimization Algorithm

AOA is a new population-based meta-heuristic algorithm proposed by et al [23]. AOA is inspired by the distribution behavior of arithmetic operators (i.e., division, multiplication, addition, and subtraction) in solving arithmetic problems. These operators are the basic elements of number theory and are used as fundamental calculation measures to examine different arithmetic problems. The optimization of complex arithmetic problems using AOA starts with the randomly generated candidate solutions, which are evaluated in each iteration using the fitness function to come closer to the best or nearly optimal solution. The two key features of the AOA algorithm are diversification and intensification search phases, which are conditioned by a function called the Math Optimizer Accelerated (MOA) function. The value of the MOA is linearly increasing from 0.2 to 0.9. The diversification phase also called exploration phase of AOA algorithm explores more possibilities and tries to cover diverse potential

solutions within the search space. The diversification phase includes Division and Multiplication search strategies. The diversification search strategies have high dispersion values that slow down the convergence process toward the target. The intensification phase also called exploitation phase, analyzes the wide range of potential solutions found in the diversification phase to come closer to the desired target solution. The intensification search phase consists of Addition and Subtraction search strategies. It have lower dispersion values that enable the operators to easily approach the target. Each candidate solution is extensively investigated using a diversification search phase when $rand_1 > MOA$ ($rand_1$ is a random value), in the case of $rand_1 < MOA$, the candidate solution converges towards an optimal solution by employing an intensification search phase. MOA is valuable function that achieves an equilibrium state between the search phases. This enable the AOA to converge towards the optimal solution. The algorithm stops once the iteration reaches to maximum iteration. The next section modeled the distribution behavior of AOA for cluster optimality.

## Proposed solution

In this section, I present the niceties of the proposed system paradigm which exhibit the use of the AOA algorithm to optimize the number of clusters. The system model of the proposed model encompasses search solutions which are randomly generated within the search space. Each candidate solution includes a different set of secure clusters. In each iteration, the potential candidate solution is either explored or exploited to move them to the optimal or near-optimal solution. The best optimal solution is determined according to the objective function. To achieve the optimality of secure clusters, the proposed approach consists of four phases i.e., i) search space creation ii) secure cluster matrix creation iii) fitness matrix creation, and iv) exploration or exploitation behavior of AOA. The description of each symbol used in equations and algorithms is given in Table 1.

**Table 1. Symbols description.**

| Notations | Meaning |
|---|---|
| $It_C$ | current iteration |
| $It_M$ | maximum iteration |
| $It_N$ | next iteration |
| $\Pi$ | performance value of automobile |
| $\psi$ | Delta Difference |
| $\mu$ | distance neighbor |
| $\underline{\amalg}$ | candidate solution |
| $\delta_k$ | CH of $k^{th}$ cluster |
| $\Delta$ | Delta value |
| $\rho_{(i,\,j)}$ | $j^{th}$ location of $i^{th}$ candidate solution |
| $\gamma_k$ | cluster members of $k^{th}$ cluster |
| $\gamma_{Avg}$ | Average cluster members of $k^{th}$ CH |
| $\varrho$ | control parameter |
| $\epsilon$ | small integer number for exploration search strategy |
| $\phi_c$ | size of candidate solution |
| $\alpha$ | sensitive parameter |
| $Z_\Pi$ | pool of high performance vehicles |
| $BestSol_f$ | fitness value of best solution |
| $BestSol_{Loc}$ | location of best solution |

## Search space creation

The search space of the proposed solution is created by the random initialization of the candidate solution along with the initialization of automobiles. Within the search space, all the vehicles are connected like a graph where vertices indicate the ID of automobiles. The edges between the vertices contain the Math Optimizer Accelerated (MOA) function which direct the candidate solution to adapt either diversification search phase or intensification search phase. MOA is updated over the course of iteration to achieve the state of equilibrium between the search phases. The MOA function is calculated as bellow:

$$\mathrm{MOA}(\mathrm{It_C}) = min + It_C \times \left( \frac{max - min}{It_M} \right) \tag{1}$$

Where $It_C$ and $It_M$ are the current iteration and maximum iteration respectively while max and min represent the maximum and minimum value of the function respectively.

## Secure cluster formation

The candidate solution is encoded as a cluster matrix, which is created by first selecting a secure CH and then adding a secure member to the cluster. The selection of a CH is based on the performance value of each vehicle. The vehicle with the highest performance value is selected as the CH. The vehicle's performance value is based on the record of data transmission. If a vehicle transmits a fake message, it will receive a penalty of (-1) and, in the case of transmitting a legitimate message, a reward of (+1) will be assigned to the vehicle. Each vehicle's performance record is stored on the RSU. When a vehicle wants to be a member of a CHj or clusterj, it will send a request message to the CHj. The request is forwarded to the RSU for verification. RSU will inspect the vehicle attributes (i.e., vehicle ID, location, performance value, vehicle mode, and manufacturer). If the performance value of a vehicle i does not meet the threshold criteria, then the vehicle i is declared a malicious node or an unauthentic node. In another case, the vehicle is assumed to be an authentic node. After a node qualifies for cluster membership, RSU will monitor the the behavior of each member and if suspicious behavior is detected (e.g., irregular message pattern, sudden variations in performance attribute and sudden message spikes etc), RSU will temporally block the cluster member and will conduct the additional verification by looking vehicle history of data transmission.

Each message contain timestamp and crytographic nonce that help the algorithm to protect against replay attack. The timestamp refers to the time when the message was created or sent by the source node. Cryptographic nonces is a unique random number that is assign to each message exchange. All the previously used nonces are stored on the RSU for verification. When a new message is received, the CH verifies that the message timestamp fall within valid time window. If it is valid, then the message is forwarded to RSU to verify the uniqueness of the messages using crytographic nonce. If a nonce is used previously, this indicate a potential replay attack and a message will be discarded.

A profile is associated with each vehicle and consists of a vehicle id, vehicle model, vehicle location, message, and performance value.

Similarly, more secure clusters are added to the candidate solution by taking into consideration the performance value of each vehicle. While creating the candidate solution, the number of CHs must be unique and covers the entire vehicular network. To include globalization in the candidate solution, we used Roulette Wheel Selection (RWS). RWS is a method used in evolutionary algorithms, to select individual from a population according to individual's performance value. The probability of selecting an individual is proportional to performance value. RWS selects the individual with the highest probability; as a result, it minimizes local

stagnation. For instance, if a search space consists of 3 individuals with a performance value of 3,4 and 2, then the normalized probability of each individual will be 0.33, 0.44 and 0.22 respectively. Now among the three individual, RWS select the individual having 0.33 probability because this individual have highest chance of being selected. The probability of a vehicle is calculated as below:

$$P_V = \frac{\Pi_V}{sum_{(V->R)}\Pi_{(V,R)}} \tag{2}$$

Where $\Pi_V$ is the performance value of vehicle and R is the repository that consist of nodes available for clustering.

$$\coprod_{(BestCHs)} = RWS(P_V) \tag{3}$$

Where $\coprod$ is the candidate solution that contain the best clusters. When the algorithm reaches maximum iteration or covers the entire network, then it stops adding clusters to the candidate solution. Table 2 demonstrates the steps of secure cluster formation.

## Fitness function

The evaluation of each candidate solution is performed by the fitness function, where we compute the fitness value of each candidate solution. While optimizing a decision variable for vehicular clustering, we need to consider multiple objectives that need to be maximized or minimized simultaneously. The fitness function consists of two objectives because vehicular clustering is considered a multi-objective problem. The below equation is used as an

**Table 2. Pseudo code for cluster formation.**

| Algorithm 1: Cluster Formation |
|---|
| **Data**: Vehicles, Neighbor Matrix |
| **Result**: Cluster Matrix |
| **1** Procedure $CS_{(CM)}$= ClusterFormation(Vehicles, NeighborMatrix); |
| **2 while** *All nodes does not covered* **do** |
| **3**   $Veh_i \leftarrow$ Req; |
| **4**   Check the profile of $Veh_i$ i.e., Vehi $A_t = Veh_i$ ID, $Veh_i$Loc, $Veh_i$ model, $Veh_i$ performance value ($\Pi$); |
| **5**   **if** $\Pi_{Veh(i)} \geq T_{val}$ **then** |
| **6**     $\zeta_{\Pi}, \Pi_{Veh(i)}$; |
| **7**     Compute probability of all high performance vehicles P; |
| **8**     $CH_i$ = RWS(P); |
| **9**     $Neighbor_{CH}$ = find(NeighborMatrix ($CH_i$); |
| **10**    Check the profile of all neighbors i.e., Neighbors $A_t$ = N ID, N Loc, N model, N performance value ($\Pi$); |
| **11**    **if** $\Pi_{neighbor(i)} T_{val}$ **then** |
| **12**      Add neighbor as cluster member of $CH_i$; |
| **13**      Add both CHi and cluster members into candidate solution i.e., $CS_{(CM)} \leftarrow (CH_i, Cl_{member})$; |
| **14**      Rep(AllNodes)- $(CH_i, Cl_{member})$-Remove the selected CH from the clustering process; |
| **15**    **else** |
| **16**      declare neighbor(i) as malacious node; |
| **17**   **else** |
| **18**     declare Veh(i) as malicious node and cannot become CH; |
| **19** Output: $CS_{(CM)}$; |

evaluation function to analyze the search solution.

$$F = w_1 \times \psi + w_2 \times \mu \tag{4}$$

Where $\psi$ and $\mu$ are the distance neighbor and delta difference, respectively. $\psi$ is the distance between the cluster head and its members, which is computed by first obtaining the distance between the CH and its neighboring vehicles and then adding the distance values of all the clusters contained in a single candidate solution. $\psi$ is calculated below:

$$D_{(k)} = \delta_{(k)} - \gamma_{(m,k)} \tag{5}$$

$$\psi = \sum_{i=1}^{z} \left[ \sum_{m=1}^{\gamma_m} D_{(k)} \right] \tag{6}$$

Where $\delta_{(k)}$ and $\gamma_{(m, k)}$ are CH and cluster member of kth cluster respectively. To ensure the optimization of clusters, it is preferable to have a low value for $\psi$. This is because the distance between a $\delta_{(k)}$ and its neighboring vehicles have an impact on the communication protocol. When there is a large distance between the cluster head and its members, it leads to increased communication delays. As a result, reliability decreased when transmitting messages. Similarly, the fitness value for the delta difference is computed. In the VANET environment, we can set a predefined threshold value called Delta. We have chosen to set Delta to 10, which represents the number of vehicles that a CH can ideally serve. The Delta Difference refers to the value obtained by subtracting the number of cluster members in the ith cluster from Delta. Eq 7 is used to model the delta difference.

$$\mu = \sum_{i=1}^{M} |\Delta - \gamma| \tag{7}$$

where $\Delta$ is the delta. We calculate the delta difference value for each cluster of a candidate solution. It is required to have a low value of $\mu$ to escalate the scalability and communication efficiency. For example, a candidate solution i have three clusters with 7, 8, and 9 cluster members, and solution j have four clusters with 6, 7, 8, and 3 cluster members. The delta difference value for solution i is 6 ((10–7)+(10–8)+(10–9)) while for solution j it is 16 ((10–6)+(10–7)+(10–8)+(10–3)). So in this scenario, solution i is considered the optimal solution. After computing the values for delta difference and distance neighbor, we add both values and create an objective matrix that contains the fitness values for all candidate solutions. Within the objective matrix, the search solution with a minimum fitness value is deliberated as optimal search solution that contains the optimal number of CHs. Table 3 shows the steps for the objective matrix creation.

## Repositioning strategy

Each candidate solution renews its position according to the best-obtained solution. The repositioning of the candidate solution is performed using the arithmetic operators, i.e., division D, multiplication M, addition A, and subtraction S. The arithmetic operators direct the search solutions to estimate the feasible locations near the optimal solution. The feasible locations around the best solution in the current iteration are obtained by a combination of exploration and exploitation search strategy. MOA function is used to choose between the two strategies. The pseudo code of the repositioning strategy is given in Table 4.

**Exploration search strategy.** In arithmetic operations, it is observed that the division (D) and multiplication (M) operators have high dispersion and distributed values that restrict the

**Table 3. Pseudo code for objective matrix creation.**

| **Algorithm 2**: objective matrix creation |
| --- |
| **Data**: cluster matrix |
| **Result**: objective matrix |
| **1** Procedure $Obj_{Matrix}$= FitnessFunction($CS_{(CM)}$); |
| **2** $BestSol_f$ = inf **for** *i = 1 to sizeof*($CS_{(CM)}$) **do** |
| **3** Compute $\psi$ i.e., distance neighbor; |
| **4** Computer $\mu$ i.e., delta difference; |
| **5** $CS(i)_{(f)} = \psi + \mu$; |
| **6** **if** $CS(i)_{(f)} < BestSol_f$ **then** |
| **7** $BestSol_f = CS(i)_{(f)}$; |
| **8** $BestSol_{(Loc)} = CS(i)_{(Loc)}$; |
| **9** End Procedure; |
| **10** Output: $BestSol_f$ and $BestSol_{(Loc)}$; |

operators from approaching the target. Hence, the D and M operators are considered exploration operators that are used to detect near-optimal solutions via random exploration of the various prominent regions of the search space. The exploration search strategy is conditioned by the MOA function, i.e., the random number r1 must be greater than the value of MOA to perform the exploration search strategy (r1 ≥ MOA). MOA is calculated using 1. The D and M searching strategies are conditioned by another random r2, i.e., for r2 ≤ 0.5, the D search strategy will perform its operation, and in another case, the M search strategy will perform its operation instead of D. It is very necessary to consider stochastic scaling coefficients to produce more exploratory behavior and explore the various regions of the search space for feasible solutions (i.e., the number of CHs). Stochastic scaling co-efficient promotes diversity in the candidate solution by introducing randomness in the scaling process. Sometime, same type of

**Table 4. Pseudo code of repositioning strategy.**

| **Algorithm 3**: Repositioning strategy |
| --- |
| **Data**: $BestSol_f, BestSol_{(Loc)}$ |
| **Result**: Updated Candidate solution |
| **1** Procedure $CS_{Nloc}$= Update Positions ($BestSol_f, BestSol_{(Loc)}$); |
| **2** Compute the value of MOA and MOP functions; |
| **3** Generate three random values between [0 1] i.e., $rand_1$, $rand_2$, $rand_3$; |
| **4 if** $rand_1 \leq MOA$ **then** |
| **5** Use exploitation search strategy to reposition candidate location; |
| **6** **if** *rand3 ≤ 0.5* **then** |
| **7** Use '+' operator to update the $i^{th}$ solution location; |
| **8** **else** |
| **9** Use '-' operator to update the $i^{th}$ solution location; |
| **10 else** |
| **11** use exploration search strategy to reposition solution location; |
| **12** **if** *rand$_2$ ≤ 0.5* **then** |
| **13** Use ×$math operator to update$$i^{th}$ solution location; |
| **14** **else** |
| **15** Use '÷' math operator to update $i^{th}$ solution location; |
| **16** End Procedure Output: New Candidate solution; |

changes in the candidate solution, may cause the algorithm to converge earlier. Therefore stochastic co-efficient is used to avoid premature convergence. moreover, the co-efficient also help the algorithm to maintain a balance between exploration and exploitation search p. The below equation is used to model the exploration search strategy:

$$\rho_{(i,j)}It_N \quad = \begin{cases} best(\rho_j) \div (MOP + \epsilon) \times ((UB_J - LB_j) \times \varrho + LB_j) & \text{if } r_2 \leq 0.5 \\ best(\rho_j) \times MOP \times ((UB_J - LB_j) \times \varrho + LB_j) & \text{if } x \geq 0.5 \end{cases} \tag{8}$$

Where $\rho_{(i, j)}It_N$ represents the $j^{th}$ location of $i^{th}$ solution in the next iteration, $best(\rho_j)$ indicates the $j^{th}$ location of the best solution obtained so far, $\epsilon$ is a small integer number, $\varrho$ is a control parameter that is used to adjust the search strategy in the search space to avoid local stagnation, upper and lower bound of the $j^{th}$ position is represented by $LB_j$ and $UB_j$ respectively. MOP is a math optimizer probability which is calculated as below:

$$MOP(It_C) = 1 - \frac{(It_C)^{\frac{1}{\alpha}}}{(It_M)^{\frac{1}{\alpha}}} \tag{9}$$

Where $It_C$ and $It_M$ are the maximum iteration and current iteration respectively. $\alpha$ is a sensitive parameter that is used to escalate the exploitative accuracy of the searching process.

**Exploitation search strategy.** The exploitation search phase performs a deep investigation of the various prominent regions found in the exploration search phase to obtain a near-optimal solution. In arithmetic operators, subtraction (S) and addition (A) operators are considered exploitative operators because they have high-density results and can easily approach the target. The exploitation searching phase is performed when the value of the random number is less than the MOA function value (r1 ≤ MOA). The operation of the exploitative operators (S and A) is conditioned by the random number r3, i.e., in the case of r3 ≤ 0.5, the S operator performs its operation; otherwise, the A operator will perform its operation instead of S. To avoid a local optimal solution and provide diversity in the candidate solution, it is important to adjust the control parameter's value according to the scenario of the search process. The below equation models the exploitative search phase:

$$\rho_{(i,j)}It_N \quad = \begin{cases} best(\rho_j) - MOP \times ((UB_J - LB_j) \times \varrho + LB_j) & \text{if } r_2 < 0.5 \\ best(\rho_j) + MOP \times ((UB_J - LB_j) \times \varrho + LB_j) & \text{if } r_2 \geq 0.5 \end{cases} \tag{10}$$

## Pseudo code of AOACNET

Table 5 contains the proposed AOACNET algorithm. The inputs for the algorithm consist of nodes, simulation area, transmission range, and candidate solutions. Line # 1–3 creates the search space that contain the search solutions along with the initialization of the automobiles and all prerequisite parameters like a (sensitive parameter), and $\varrho$ (a control parameter used to adjust the search strategy in the search space to avoid local stagnation). Line # 4 computes the neighbor matrix using the standard Euclidean distance which computes distance between two points in the two dimension problem space. Line # 6–9 initialize the cluster matrix, the fitness value, and the location of the best candidate solution to empty. Line # 10–13 creates a cluster matrix that contains secure clusters. The size of the cluster matrix represents the total number of candidate solutions. Due to stochastic behavior and different location of a search solution within the search space, each candidate solution may contain a different number of secure clusters. The security of clusters is achieved using the performance value which is based on the record of data transmission.

**Table 5. Pseudo code of AOACNET algorithm.**

| Algorithm 4: AOACNET Algorithm |
|---|
| **Data**: Population size, number of vehicles, simulation area, transmission range |
| **Result**: optimal candidate solution, i.e., optimal secure clusters |
| **1** Create a search space and randomly initialize the location of candidate solution and automobiles; |
| **2** Initialize the direction and speed of automobiles; |
| **3** Create a mesh topology among the vehicles; |
| **4** Create the neighbor's matrix i.e., NeighborMatrix using the distance between vehicles; |
| **5** Initialize pre-requisite parameters of AOA i.e., $\varrho$ (control parameter), $\alpha$ (sensitive parameter) and $\epsilon$ (integer number); |
| **6 while** $It_C < It_M$ **do** |
| **7**    $CS_{(CM)}$ = empty // Initialize the cluster matrix to empty; |
| **8**    $BestSol_f$ = inf $BestSol_{Loc}$ = 0,0 // Initialize the fitness value to infinity and location of best solution to (0,0) |
| **9**    **for** $i=1$ to solution **do** |
| **10**        $CS_{(CM)}$ = ClusterFormation(Vehicles, NeighborMatrix) // create cluster matrix using algorithm 2; |
| **11**        ObjMatrix = FitnessFunction(CS(CM)) // create objective matrix using algorithm 3; |
| **12**    **for** $m = 1$ to solution **do** |
| **13**        $CS_{Nloc}$ = Update Positions ($BestSol_f$ $BestSol_{(Loc)}$ // Update the position of candidate solution using algorithm 4; |
| **14**    Update the fitness value of best candidate solution; |
| **15**    BestCS = $BestSol_f$; |
| **16**    $It_C$ + +; |
| **17** Output: Best Optimal CHs = BestCS |

A vehicle with a good record of data transmission have a high performance value, and a vehicle with a bad record of data transmission have a low performance value. Line # 12 evaluates the candidate solution using the fitness function. Line # 14–16 uses the distribution behavior of arithmetic operators to find the near-optimal location around the best candidate solution obtained so far. Line # 17–18 update the best candidate solution and its fitness value. After the final cycle, the algorithm returns the best search solution that have the optimal number of secure vehicular clusters, as shown in line # 21. The flowchart of the AOACNET clustering algorithm is given in Fig 4.

## Experimental setup and results discussion

We have used MATLAB 2021a on a laptop with 8GB of RAM and a 2.5 GHz of Intel core-i5 processor to implement the proposed algorithm. The system have 64 bit architecture. Different experiments are performed by varying the simulation parameters as given in Table 6. The simulation parameters are used to evaluate the performance metrics such as number of clusters and load on each cluster and computational time. The number of clusters indicate the size of individual candidate solution and load on cluster indicate the size of a single cluster within the candidate solution. The computational time refers to the time taken by the algorithm to authenticate a single entity. Each simulation parameter plays an important role in finding the optimal result like control parameter is used to avoid local optima, sensitive parameter that escalate the searching process, transmission range and network size etc. A comparison of the proposed algorithm have been made with the benchmark competitors i.e., CLPSO, GWOC-NET, and CACONET. Results exhibit the efficacy of the proposed solution in obtaining secure optimal clusters. This results in the substantial performance boost of the vehicular network by attaining elevated levels of reliability and scalability. In addition, the optimal cluster also

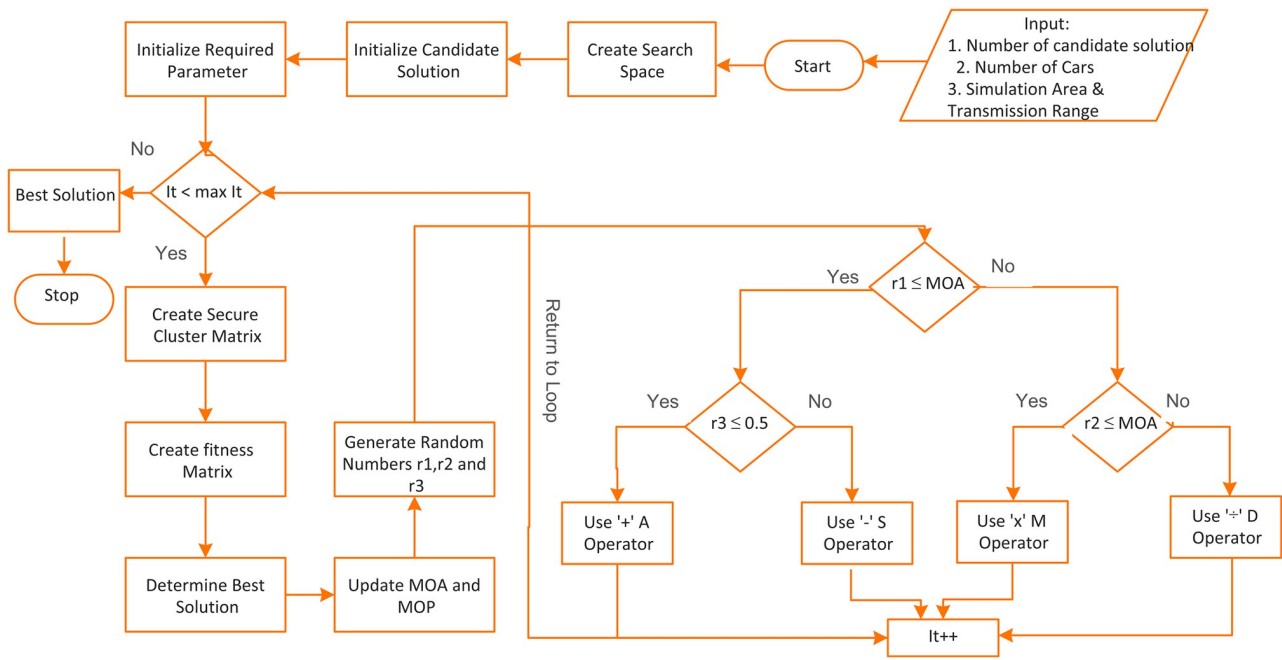

**Fig 4. Flowchart of proposed algorithm.**

reduces communication latency and network overhead which is created due to the formation and maintenance of too many clusters.

## Results discussion

The proposed algorithm is tested against similar approaches using two performance metrics, i.e., the number of secure clusters and load balancing. Load balancing indicates the load on

**Table 6. Simulation parameters.**

| Parameters | AOACNET |
| --- | --- |
| Transmission Range | (100–600)m |
| Grid Size | (1x1–2x2)km |
| Nodes | 30–60 |
| Control parameter ($\varrho$) | 0.5 |
| candidate solution | 100 |
| dimention | 2 |
| number of lane | 4 |
| lane size | 40 |
| iteration | 1–150 |
| weights | 0.5 |
| velocity limit | $(25–30)ms^{-1}$ |
| sensitive parameter $\alpha$ | 5 |
| $\Delta$ | 10 |
| $T_p$ | 1.7ms |
| Packet Data Size | 512 bytes |
| RSU Transmission Power | 1000m |
| $T_{trans}$ | 0.5ms |
| simulation run | 10 |

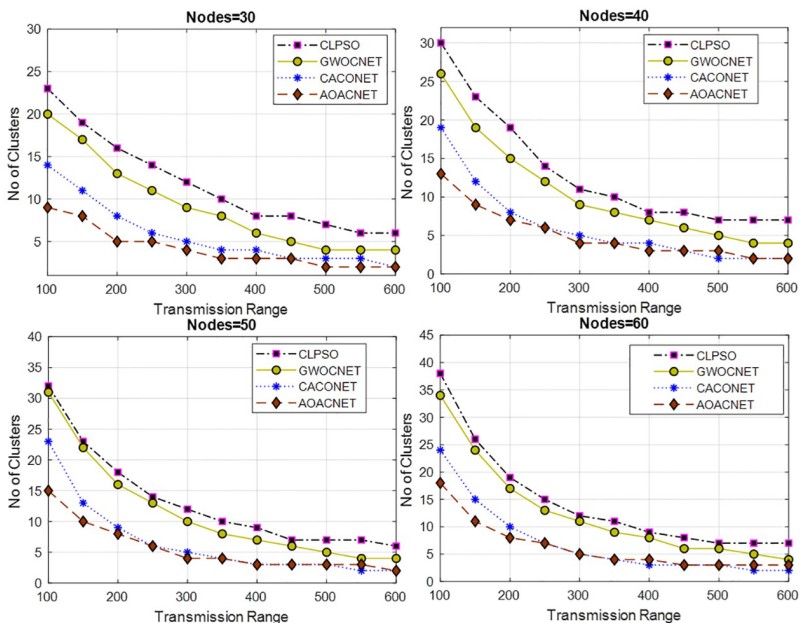

**Fig 5. Transmission range versus secure clusters and grid size = 1x1Km.**

each cluster, which is computed by considering the ratio between the actual and obtained neighboring vehicles of cluster during the period of simulation. The performance metrics are studied by varying simulation parameters such as simulation area, network size, and transmission range etc.

**Secure cluster versus transmission range.** This experimental setup is performed in a simulation area of 1x1 km and 2x2 km with 30 to 60 vehicles having a transmission range of 100m to 600m. For each set of vehicles, various experiments are performed against the transmission range to obtain secure clusters. The results in Figs 5 and 6 reveal that the proposed solution offers efficient performance as compared to other approaches.

The proposed solution is able to provide a feasible solution to cover the vehicular network with an optimum dominant set of secure clusters that results in the efficient utilization of scarce bandwidth resources. We can see that there is a strong direct relationship between the transmission range and the number of secure clusters. Upon increasing the transmission range, the total set of secure clusters drops because the coverage area of each CH increases, which covers more members in its purlieu.

It is noted that increasing the simulation area also increases the set of secure clusters because the vehicles are deployed far away from each other. In each scenario, the proposed algorithm performs better in plummeting the communication hops, resulting in the minimization of communication latency. Moreover, this also minimizes the overhead on the network because fewer resources are required to lodge the clusters. Due to secure clusters, the chance of malfunctioning the security and privacy of the vehicular network is also reduced. Consequently, this also increases the message's accuracy and attains an elevated level of data reliability.

**Secure clusters versus vehicular nodes.** This experimental framework is performed in a simulation area 1x1 km and 2x2 km for a transmission range of 100m to 400m with a varying number of vehicular nodes i.e., 30 to 60. In each experiment, the dominant set of secure clusters is obtained against the varying sets of vehicles while keeping the transmission range

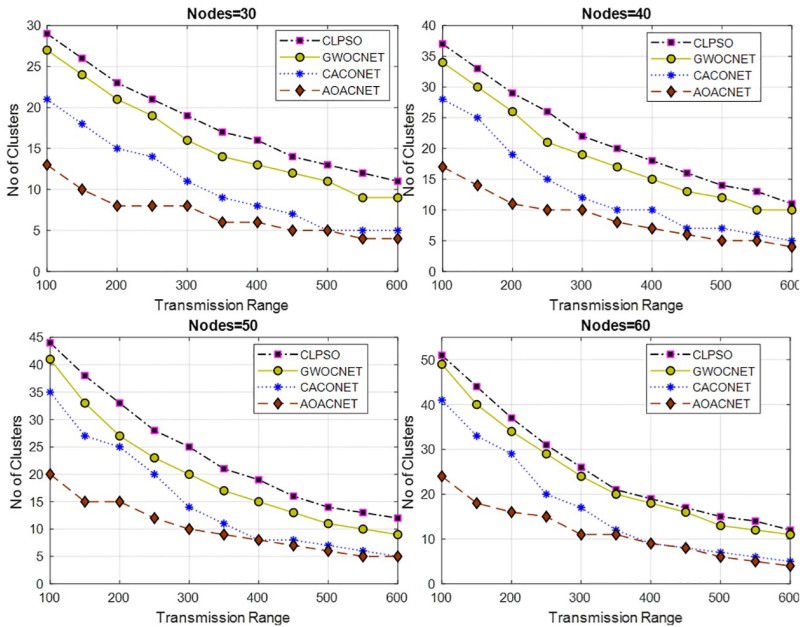

**Fig 6. Transmission range versus secure clusters and grid size = 2x2Km.**

constant. The results signify the efficient fallout of the proposed approach as compared to similar competitors. The simulation results in Figs 7 and 8, reveals that the network congestion in terms of nodes can strongly affect the operation of the protocol. In this situation, the performance of similar competitors is almost the same in terms of optimal clusters but the proposed algorithm still performs better even in the large densities of vehicles by obtaining optimum

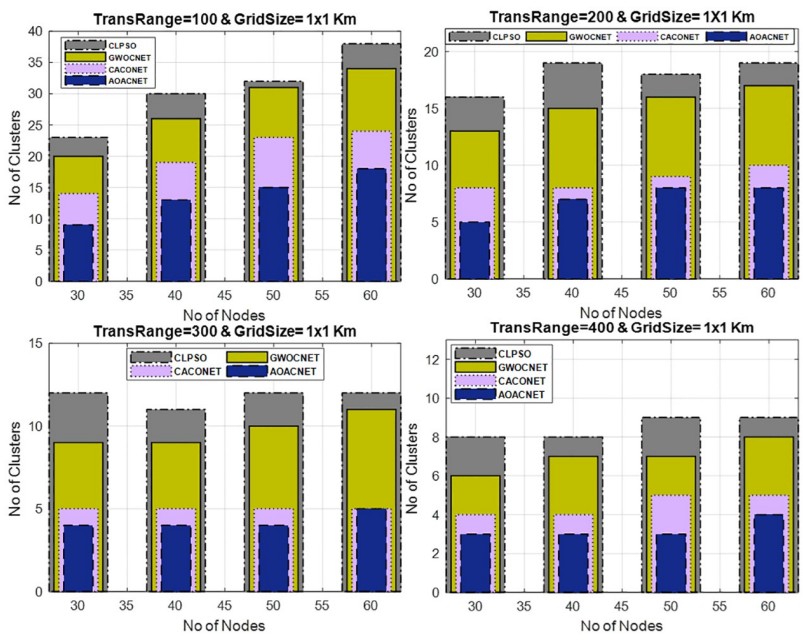

**Fig 7. Vehicular node versus secure clusters and grid size = 1x1Km.**

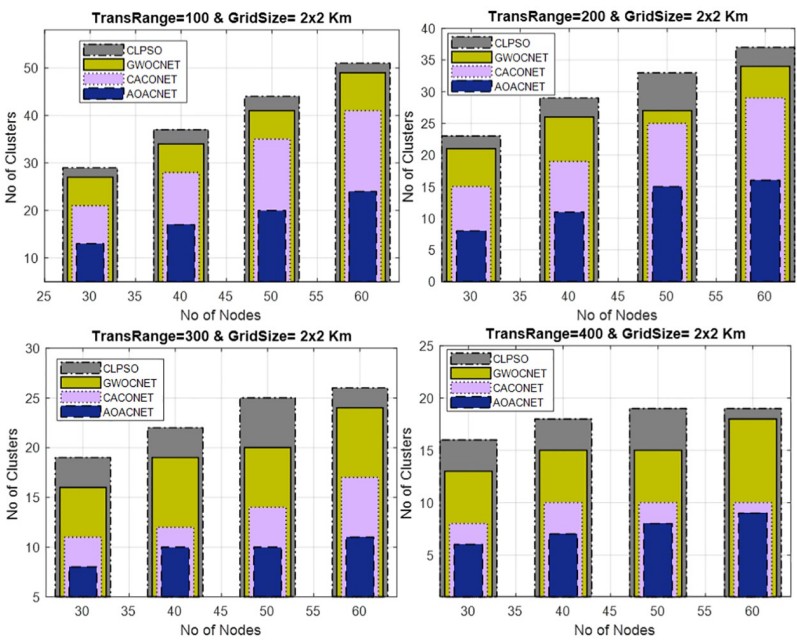

**Fig 8. Vehicular node versus secure clusters and grid size = 2x2Km.**

secure vehicular clusters. Consequently, this improves the scalability and robustness of the vehicular network.

**Secure clusters versus network area.**  This experimentation is performed by varying the network area while keeping the network vehicles and transmission range constant. The results are generated for a set of 40 vehicles, and the transmission range of each vehicle is set to 300m to 600m. Again, we can see the efficient fallout of the proposed approach in Fig 9 as compared to other algorithms.

As a whole, we can safely infer that the proposed approach is a stable and useful algorithm to tackle the combinatorial optimization problem, i.e., vehicular clustering. The reason behind the efficient performance of AOA clustering algorithm is simple and straight forward implementation of the algorithm. It have an extensive set of stochastic variables and an effective designing of control variable enables the proposed algorithm to equalize the search strategies of exploration and exploitation. The effective design of control variable produces stochastic value to maintain a balance between exploration and exploitation search strategies till the last iteration, which aid the algorithm to converge towards global optimal solution by averting local stagnation.

**Load Balance Factor (LBF).**  The load balance factor is a performance metric that depicts the load borne by the single CH. For efficient communication, it is very necessary to balance the load on each cluster. However, due to the fast mobility of cars, attaining equal load balancing is almost impractical. So it is desirable to measure the load on each cluster to analyze the performance of the proposed algorithm. The Eq 11 is used to quantify the load on a CH.

$$l = \frac{1}{\left(\phi_c \times \sum_{k=1}(\gamma_k - \gamma_{Avg})\right)} \tag{11}$$

$\phi_c$ indicates the size of the candidate solution, and $\gamma_k$ indicates the number of cluster members in the $k^{th}$ cluster head. $\gamma_{Avg}$ represents the average cluster members of cluster head and is

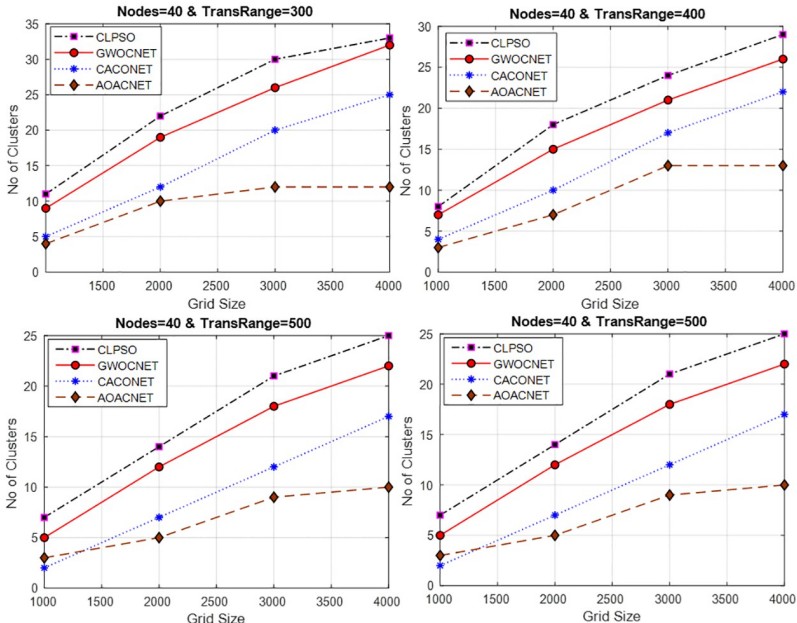

**Fig 9. Network area versus secure clusters and transmission range = 300–600m.**

calculated as below:

$$\gamma_{Avg} = \frac{(N - \phi_c)}{\phi_c} \tag{12}$$

It is desirable to have a high value of l to achieve better distribution of vehicles among cluster heads. To compute the value of l, an experiment in a simulation area of 1x1 and 2x2 km is conducted while the transmission range and network vehicles are set to 100m to 600m, 30, and 60, respectively. The results in Fig 10 reveal the efficient performance of the proposed clustering algorithm in terms of the efficient distribution of vehicular nodes to cluster heads as compared to the benchmark approaches. This indicates that the proposed approach balances the load among all the clusters as compared to other approaches. The numerical analysis of the proposed algorithm is given in Table 7. The table have 5 columns, where columns 2 and 3 represent the percentage of clusters selected in each network area. Columns 4 and 5 represent the percentage of cluster heads and cluster members for all network areas. Hence, we can conclude from the numerical data of the proposed algorithm that AOACNET outperforms the competitors in optimizing the number of secure vehicular clusters by up to 25%.

## Security analysis

In this section, we analyze the security of proposed algorithm against malicious node detection and malicious packet detection and prevention. It is important to analyse the security of the proposed algorithm to identify potential vulnerabilities that may exploit by the malicious nodes. The malicious nodes can harm the network in various ways such as increase response time and transmission latency, forwards duplicates packets to increase overhead on the network. Moreover, malicious node reduce trust level among vehicles that generate serious issues like loss of precious human life and damage properties.

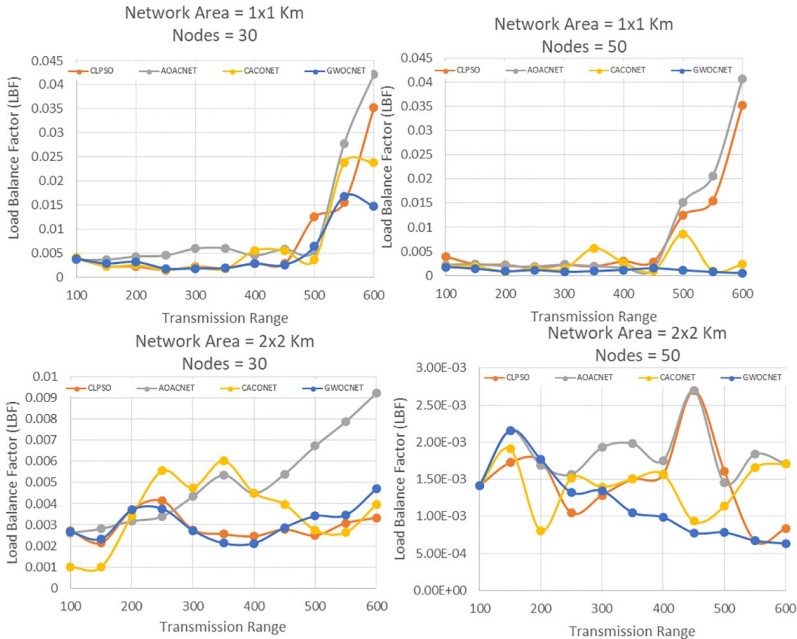

**Fig 10. Transmission range versus LBF, network area = 1x1–2x2km and nodes = 30 and 50.**

The security of proposed algorithm is analyzed against various attacks like detection of impersonation attack, detection of bogus messages attack, detection of replay attack and Sybil attack detection.

**Detection of impersonation attack.** *Impersonation attack*. In this attack, the deceiver pretends to be a legitimate node to join the cluster.

*Resistance to impersonation attack*. To perform the impersonation attack, the attacker must have a higher performance value than the defined threshold. The proposed algorithm employs performance attributes that rely on the history of data transmission. This implies that a legitimate member of a cluster have a high-performance value to transmit legitimate messages i.e., $\Pi_V > T_v$. A performance attribute is updated with time as a result of their behavior and transmission. A key factor that acts as a deterrent against impersonation attacks is the penalty (-1) for transmitting fake messages. A malicious node attempting impersonation by consistently transmitting fake messages receives more penalties. As penalties accumulate, the performance value of the malicious vehicle will become negative i.e., $\Pi_V < T_v$. This violates the eligibility condition for becoming a legitimate vehicle. Thus preventing vehicles from joining the cluster.

Moreover, the performance attribute ($\Pi_V$) is dynamically adjusted which enable the proposed algorithm to adapt to node behavior changes. A decreasing performance value of vehicles attempting impersonation attack, making it complicated for them to maintain legitimacy. Thus the proposed algorithm is efficient in detecting the impersonation attack.

**Table 7. Numerical analysis.**

| Algorithm | Simulation Area | % of CH | % of cluster member |
|-----------|-----------------|---------|---------------------|
| AOACNET | 12.23%, 21.34% | 16.78% | 83.21% |
| CACONET | 15.03%, 31.87% | 23.45% | 76.92% |
| GWOCNET | 24.98%, 45.95% | 35.07% | 64.92% |
| CLPSO | 30.46%, 51.73% | 41.10% | 58.89% |

**Detection of bogus messages attack.**   *Resistance to bogus message attack*. To send a malicious or bogus message, an attacker needs a higher packet attribute value than the defined threshold. However, it is very difficult for the deceiver to maintain a high packet attribute value due to its suspicious and irregular data patterns. The proposed algorithm uses a packet profile that is dynamically adjusted based on network conditions, previous data patterns, and security threats. Moreover, the proposed algorithm employs a two-tier verification process (CH and RSU) to comprehensively analyze the packet profile for the detection of bogus messages, making it challenging for attackers to propagate bogus message attacks within the network. The successful verification of a bogus message attack will result in discarding the message to prevent the propagation of the attack and to safeguard the integrity of the network.

**Detection of replay attack.**   *Replay attack*. In a replay attack, the attacker intercepts and re-inject the previously received packets into the network without actual interactions with the peers.

*Resistance to replay attack*. To protect against replay attacks, the proposed algorithm employs timestamp along with cryptographic nonces. The timestamp is used to verify that the message is received within a valid time window while the cryptographic nonce (a unique number for each message exchange) verifies that the message carries a unique nonce. If a message contains previously used nonce, this indicates a potential replay attack. Hence, the proposed algorithm is efficiently able to detect replay attacks.

**Detection of Sybil attack.**   *Sybil attack*. In a Sybil attack, the attacker creates multiple malicious nodes to gain control of a network.

*Resistance to Sybil attack*. A Sybil attacker infiltrates the system by injecting various Sybil nodes (malicious nodes) with the intent to disturb the normal operation of the network by transmitting irrelevant messages. The proposed algorithm can detect bogus messages via performance attributes. Therefore, when a Sybil node transmits a fake message, it will receive a penalty of (-1) to its performance value i.e., $\Pi_V = \Pi_V - 1$. After transmitting N bogus messages, the net change in the performance value is $\Delta\Pi_V = -N$. For a successful infiltration into the system, the performance value of a Sybil node must be greater than the threshold i.e., $\Pi_V + \Delta\Pi_V > T_v$. However, receiving penalties for transmitting fake messages, makes it increasingly challenging for the Sybil node to maintain high performance value. Thus the proposed algorithm is efficient in detecting the Sybil node attack.

**Detection of node hijacking.**   *Node hijacking*. This involves the legitimate node being controlled by a malicious node to carry out malicious activities.

*Resistance to node hijacking*. The proposed algorithm efficiently responds to node hijacking attacks through a continuous and real-time monitoring system which is particularly effective in detecting such attacks. After selecting a node as a cluster member, if its behavior becomes inconsistent with its historical data (e.g., sending irregular message patterns at unusual times, sudden variation in performance value, sending high volumes of data, RSU flags this node as suspicious. The RSU then initiates the additional verification process, including checking the node's history and message patterns. This process helps the proposed algorithm in detecting node hijacking attacks.

**Computation time of authentication process.**   The efficiency of the proposed scheme is compared with other existing security schemes in terms of the computation time for authenticating one or several vehicles. To compute the computation cost, we consider $T_p$ is the time taken for process the authentication request. This includes various operations like comparison of performance value with threshold, verification of message validity using a timestamp, and verification of message cryptographic nonce and $T_{trans}$ is the transmission time.

For simulations, the processing time $T_p$ and transmission time $T_{trans}$ are set to 1.7ms (milliseconds) and 0.5ms respectively. The simulation is conducted in a network environment,

**Table 8. Computation time of authenticating vehicles of different algorithms.**

| Algorithm | Authentication one vehicle | Authentication of n vehicle |
|---|---|---|
| Sun's Algorithm | $7T_p + T_{trans}$ | $7nT_p + nT_{trans}$ |
| Khan's Algorithm | $11T_p + T_{trans}$ | $11nT_p + nT_{trans}$ |
| Subramani's Algorithm | $6T_p + T_{trans}$ | $6nT_p + nT_{trans}$ |
| Yanwei's Algorithm | $5.5T_p + T_{trans}$ | $5.5nT_p + nT_{trans}$ |
| Proposed Scheme | $4T_p + T_{trans}$ | $4nT_p + nT_{trans}$ |

where vehicles communicate with each other and with the RSU within a 1x1 km simulation area. The computation time is recorded for maximum number of nodes i.e, 60. In Table 8, we compute the computation time of the proposed scheme and other schemes. We can see that the proposed algorithm consumes less time to authenticate a vehicle for membership of a cluster i.e., $4T_p + T_{trans}$.

As shown in Fig 11, the proposed algorithms take 7.3 ms to authenticate a single node whereas Sun's and Khan's take 12.4 ms and 19.2 ms respectively. Thus it is understandable that the proposed algorithm is computationally efficient as compared to other schemes.

**End-to-end delay.** In this section, we evaluate the performance of the secure clustering algorithm in terms of end-to-end delay. Various experiments are conducted over a varying transmission range in a simulation area of 1x1 km while keeping the number of nodes constant i.e., 60. Other simulation parameters i.e., packet data size, vehicles speed and RSU transmission power are kept constant. End-to-end delay is calculated by taking into consideration the processing delay, transmission delay, detection delay, and delay that occurs due to a malicious node. The results in Fig 12 demonstrate the efficacy of the proposed algorithm in obtaining minimum end-to-end delay as compared to benchmark algorithm. The comparative algorithms incur high end-to-end delays because they didn't consider any security measures to protect them from malicious vehicles. Malicious vehicles can disturb the operation of algorithms such as manipulate routing information, that result in the non-optimal routing paths. As a result, the packet follows the longest path towards the destination. Malicious vehicle may

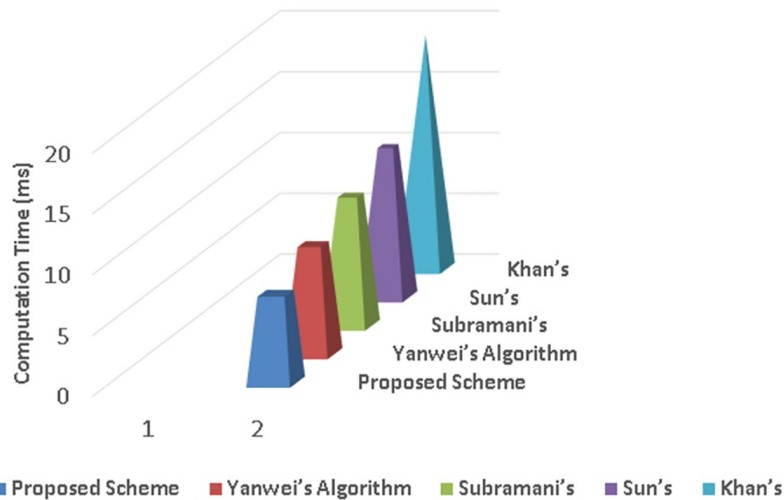

**Fig 11. Comparison of computational time of proposed versus other schemes.**

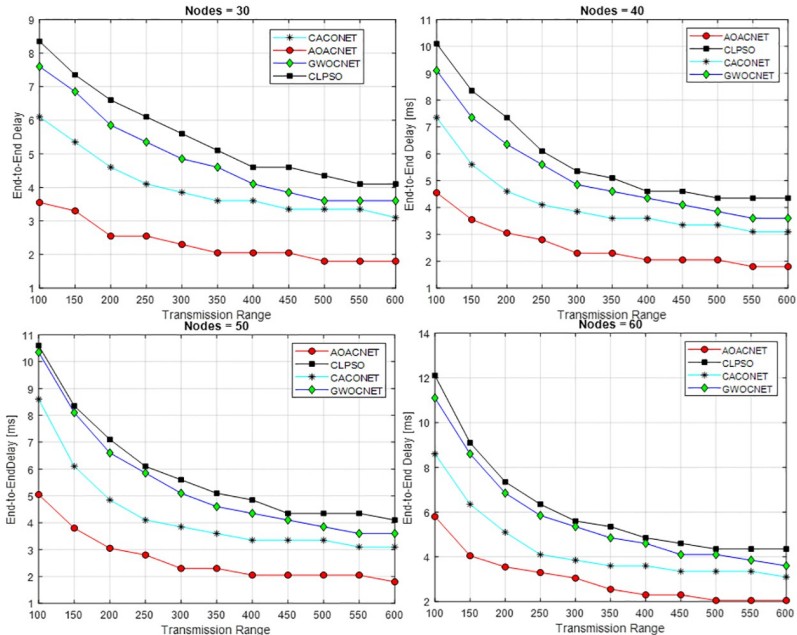

**Fig 12. End-to-end delay versus transmission range in simulation area = 1x1 km.**

generate duplicate packets to increase network congestion. This increases the packet drop ratio and requires source to re-transmit each packet. As a result end-to-end delay increases.

**Network overhead.** This framework analyzes the security of the proposed algorithm in terms of network overhead. The results are obtained by varying transmission ranges in a simulation area of 1x1 km km while keeping the rest of the simulation parameter (e.g., packet data size, RSU transmission power etc) constant. The The network overhead is calculated by taking into consideration the overhead that occurs due to the number of cluster formations, inter-cluster communication, intra-cluster communication, detection overhead, and overhead due to malicious attacks. The performance attribute of each vehicle aids the algorithm in detecting malicious nodes. The result in Fig 13 reveals the efficient performance of the proposed algorithm in producing minimum overhead on the network as compared to the benchmark algorithms. Insecure clusters are vulnerable to various attacks, such as DoS attacks. These attacks can increase communication overhead and increase the consumption of network resources. The attacks can interfere with legitimate communication to disturb the operation of the protocol by injecting fake messages into the network. As a result, the network overhead increases.

## Conclusion

In this work, we explore vehicular clustering as combinatorial optimization problem using the Arithmetic Optimization Algorithm (AOA) to address the issue of the scalability and reliability of the vehicular network. To this end, the proposed algorithm imitates the distribution behavior of arithmetic operators to optimize secure vehicular clusters. Vehicles are permitted to join clusters by evaluating its performance value which is dynamically adjusted based on the legitimacy of their transmitted messages. The algorithm starts its operation with the initialization of search solution and automobiles within the search space. In each cycle, the candidate solution is either explored or exploited using the exploration search and the exploitation search strategy

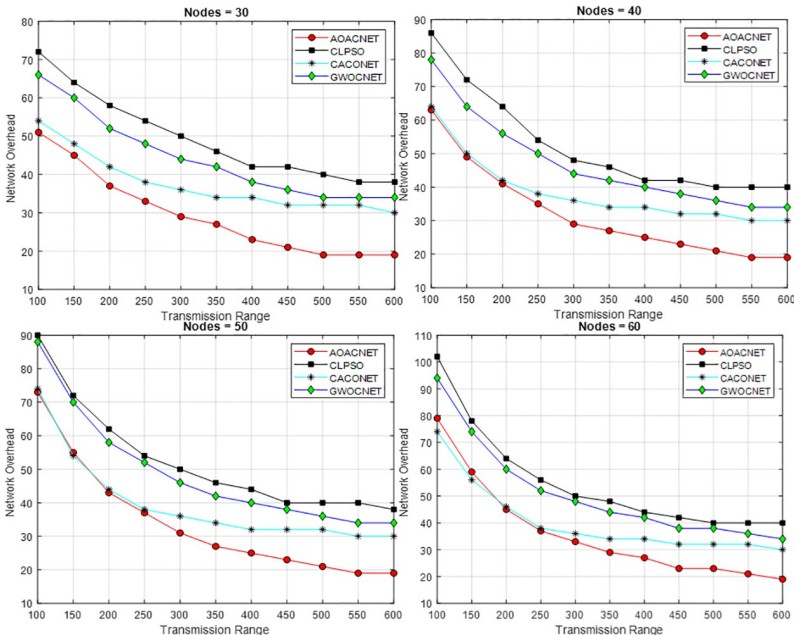

**Fig 13. Network overhead versus transmission range in simulation area = 1x1 km.**

to get the best possible solutions, which are further evaluated using the evaluation function to obtain the best solution, i.e., secure optimal clusters.

Extensive simulation in MATLAB demonstrate the efficacy of AOACNET up to 25% in terms of designated optimization objectives. It implies that the proposed algorithm minimizes the communication hops between source and destination, which results in a reduction of communication latency and communication overhead. Moreover, due to secure clusters, the risk of different malicious attacks is minimized, which improves vehicular networks user privacy and escalate transmission reliability and accuracy. The optimization of vehicular clusters increases transportation efficiency by minimizing data collisions and ensuring the timely delivery of safety data packets. The proposed clustering algorithm can adapt to emerging technologies by considering the unique challenges faced by 5G and 6G communication architectures. 5G and 6G communication architectures introduce high data rates and have low latency requirements. To meet the strengthen requirements of 5G and 6G communication architecture, we can explore the use of edge computing to offload the computational tasks from RSU to edge servers. This will minimize the communication latency and improves performance in a densely populated vehicular network.

Moreover, We can extend the proposed algorithm by incorporating machine learning techniques such as deep reinforcement learning to identify traffic patterns, vehicle behavior and network conditions to make adaptive decisions. Using deep learning, the algorithm can learn from past experiences to dynamically adjust performance value threshold for cluster membership based on real-time network conditions. This integration improves the accuracy in detecting malicious attacks, reduces communication delay and network overhead.

## Supporting information

**S1 Data.**
(RAR)

## Author Contributions

**Conceptualization:** Asad Ali, Muhammad Assam, Masoud Alajmi.

**Data curation:** Salgozha Indira, Hend Khalid Alkahtani.

**Formal analysis:** Masoud Alajmi, Tahani Jaser Alahmadi.

**Funding acquisition:** Hend Khalid Alkahtani.

**Methodology:** Asad Ali, Muhammad Assam, Yazeed Yasin Ghadi, Salgozha Indira.

**Resources:** Hend Khalid Alkahtani.

**Software:** Yazeed Yasin Ghadi, Ainur Akhmediyarova, Tahani Jaser Alahmadi.

**Writing – original draft:** Asad Ali, Muhammad Assam.

**Writing – review & editing:** Yazeed Yasin Ghadi, Ainur Akhmediyarova, Hend Khalid Alkahtani.

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
