## [Decision Letter · Decision Letter 0]

14 Jan 2024

PONE-D-23-43275Arithmetic Optimization based Secure Intelligent Clustering Algorithm for Vehicular Adhoc NetworkPLOS ONE

Dear Dr. Ali,

Thank you for submitting your manuscript to PLOS ONE. After careful consideration, we feel that it has merit but does not fully meet PLOS ONE’s publication criteria as it currently stands. Therefore, we invite you to submit a revised version of the manuscript that addresses the points raised during the review process.

We look forward to receiving your revised manuscript.

Kind regards,

Pandi Vijayakumar, Ph.D

Academic Editor

PLOS ONE

Journal Requirements:

   "This work was supported by the Princess Nourah bint Abdulrahman University Researchers Supporting Project number (PNURSP2023R384), Princess Nourah bint Abdulrahman University, Riyadh, Saudi Arabia"

5. Please upload a copy of Figures 1-10, to which you refer in your text on pages 2-23. If the figure is no longer to be included as part of the submission please remove all reference to it within the text.

Additional Editor Comments:

The paper has many serious technical issues with respect to the framework, contribution and figures. Also, the reviewers have given a lot of serious comments which must be corrected while submitting the revised version. So The authors should carefully revise the paper and submit the revised version.

Reviewers' comments:

Reviewer's Responses to Questions

**Comments to the Author**

1. Is the manuscript technically sound, and do the data support the conclusions?

Reviewer #1: No

Reviewer #2: Partly

2. Has the statistical analysis been performed appropriately and rigorously? 

Reviewer #1: No

Reviewer #2: Yes

3. Have the authors made all data underlying the findings in their manuscript fully available?

Reviewer #1: No

Reviewer #2: Yes

4. Is the manuscript presented in an intelligible fashion and written in standard English?

Reviewer #1: No

Reviewer #2: Yes

5. Review Comments to the Author

Reviewer #1: This article proposes an algorithmic solution to the vehicular clustering problem using the Arithmetic Optimization Algorithm (AOA). The AOA mimics arithmetic operators to optimize the formation of secure vehicular clusters. Through extensive MATLAB simulations, the proposed approach is compared with benchmark algorithms, demonstrating its effectiveness in enhancing vehicular network reliability, scalability, and communication efficiency. The study evaluates the algorithm's performance under varying simulation parameters, such as transmission range, network size, and area, highlighting its potential for future applications in advanced vehicular and airborne network scenarios.

The article lacks proper organization with the absence of headings and subheadings throughout its content.

Abstract:

Clarify the specific meta-heuristic algorithm employed by AOACNET.

Provide more details on the threshold condition for authenticating nodes during cluster formation.

Specify the nature of the extensive simulation results and the metrics used for evaluation.

Introduction:

Elaborate on the unique challenges posed by the dynamic network topology in VANETs.

Include a concise comparison with existing clustering methods in VANETs to emphasize the innovation of AOACNET.

The section provides a comprehensive overview of the challenges in Vehicular Adhoc Networks (VANET) and the importance of secure vehicular clustering. However, consider breaking down the lengthy paragraphs into smaller ones for improved readability.

Motivation and Problem Statement:

The motivation for secure vehicular clustering is well-stated, particularly emphasizing the impact of malicious nodes on passenger safety. Consider providing specific examples or scenarios to illustrate the severity of the problem.

Introduction to Vehicular Clustering:

The introduction to vehicular clustering is informative. However, you may want to elaborate more on the benefits and challenges of clustering in VANETs. How does clustering address the issues of dynamic network topology and fast vehicle movement?

Importance of Optimal Clustering:

The importance of optimal clustering is highlighted, touching on factors like communication efficiency and resource utilization. Consider providing more concrete examples or studies that demonstrate the impact of optimal clustering on VANET performance.

Introduction to AOA (Arithmetic Optimization Algorithm):

The introduction to AOA is clear, but you might want to briefly explain why AOA is chosen over other meta-heuristic algorithms. What specific characteristics of AOA make it suitable for vehicular clustering?

Clarity and Organization:

The literature review provides a comprehensive overview of clustering protocols in vehicular networks. However, consider breaking down the section into smaller subsections for better organization, making it easier for readers to navigate through the content.

Taxonomy of Clustering Protocols:

The classification of clustering protocols into general-purpose and domain-specific categories is clear. Consider providing a brief summary or transition statement before introducing Fig 3 to prepare the reader for the taxonomy.

Fig 3 - Classification of Clustering Protocols:

The taxonomy figure is a valuable addition. However, ensure that the figure is adequately explained in the text. Each category and its significance should be briefly discussed.

General Purpose Clustering Protocols:

The descriptions of general-purpose clustering algorithms are informative. Consider adding a sentence or two after each algorithm to highlight its strengths or limitations in the context of vehicular networks.

Domain-Specific Clustering Protocols:

The categorization of domain-specific clustering protocols into routing, security, MAC, and QoS is effective. Ensure that each sub-section provides a concise yet comprehensive overview of the protocols within that domain.

Routing Protocols:

The description of clustering in the context of vehicular routing is clear. Consider adding a brief sentence about the importance of clustering in improving routing efficiency in VANETs.

Security Protocols:

The overview of security protocols is well-structured. However, for each protocol, consider providing a sentence about how clustering enhances security in vehicular networks.

MAC Protocols:

The section on MAC protocols is comprehensive. Consider briefly explaining how cluster-based MAC protocols address challenges like fair channel access in vehicular networks.

QoS Assurance Protocols:

The discussion on QoS assurance protocols is well-done. Consider emphasizing the role of clustering in maintaining QoS metrics and how it contributes to the overall network performance.

Meta-Heuristic Based Clustering Algorithms:

The explanation of meta-heuristic algorithms is clear. Consider adding a sentence after each algorithm to highlight its specific advantages in the context of clustering for vehicular networks.

Arithmetic Optimization Algorithm (AOA):

The introduction of AOA is well-described. Consider adding a sentence or two about why AOA is chosen as the focus of the paper and how it differs from other meta-heuristic algorithms.

Distribution Behavior of AOA:

The explanation of the distribution behavior of AOA is clear. Consider providing a brief example or analogy to help readers conceptualize the diversification and intensification phases.

Explanation of MOA Function:

The MOA function is introduced, but there's no detailed explanation of its purpose or significance. Provide more context on how it contributes to the search space creation.

Explanation of Search Space:

Clarify how the search space is represented and how candidate solutions move within this space. A visual representation or example would be beneficial.

Secure Cluster Formation (Lines 416-443):

Explanation of Secure Cluster Formation:

The section is detailed, but consider breaking down complex processes, such as Roulette Wheel Selection (RWS), into smaller steps for better understanding.

Roulette Wheel Selection (RWS):

Provide a bit more context or a simple example of how RWS works in the context of secure cluster formation.

Fitness Function

Objective Function Explanation:

Offer a brief explanation of why a multi-objective fitness function is chosen and how ψ and μ contribute to the evaluation of solutions.

Delta Difference (Equation 7):

Provide a bit more explanation or intuition behind the delta difference concept.

Repositioning Strategy

Clarification on Exploration and Exploitation:

Clearly explain the roles of exploration and exploitation in the context of repositioning. It might help to provide an example or visual aid.

Mathematical Notation:

Ensure that all symbols used in equations are defined or referenced in Table 1 for clarity.

Exploration and Exploitation Search Strategy

Explanation of Exploration and Exploitation:

Offer more insight into why certain operators are considered explorative or exploitative and how they affect the search process.

Stochastic Scaling Coefficients:

Provide a brief explanation of why stochastic scaling coefficients are necessary and how they impact the algorithm.

Pseudo Code of AOACNET

Initialization (Lines 1-3):

Clarify the purpose of parameters like 'a' and 'ϱ' in the initialization step.

Neighbor Matrix Calculation (Line 4):

Provide more details on how the Euclidean distance is used to compute the neighbor matrix. Since there are many distance calculation methods are available example Manhattan Distance, Minkowski Distance, and Hamming Distance.

Cluster Matrix Creation (Lines 10-13):

Clarify how the candidate solution contains a different number of secure clusters.

Distribution Behavior of Arithmetic Operators (Lines 14-16):

Offer more details on how the distribution behavior of operators is determined and its significance in finding near-optimal locations.

Final Output (Line 21):

Specify what the final output represents—is it the best solution in terms of the fitness function?

Consider adding comments within the pseudo-code for better readability.

Provide more contextual examples or figures where applicable to aid understanding.

Ensure that all symbols, variables, and parameters are consistently defined and explained.

Review the algorithm for any potential edge cases or scenarios not covered in the explanation.

Consider including a summary or conclusion at the end of the section to reinforce key points.

MATLAB Version:

Specify the MATLAB toolbox or specific functions used in the implementation. This adds transparency and helps others replicate the results.

Experimental Setup (Lines 538-547):

Hardware Specifications:

Include more details about the laptop's specifications, such as the processor model and architecture, to provide a clearer understanding of the computational resources.

Simulation Parameters (Table 6):

Clearly explain the significance and impact of each simulation parameter on the results. This will help readers understand the experimental design better.

AOACNET Algorithm (Lines 519-536):

Initialization (Line 1):

Explain the significance of the "direction and speed of automobiles" in the initialization step.

Cluster Formation (Lines 9-13):

Provide more details on how the cluster matrix is initialized and how the cluster formation process contributes to the overall algorithm.

Update Positions (Lines 13-15):

Clarify how the positions are updated, and explain the role of the MOA and MOP functions in this context.

Results Discussion (Lines 548-574):

Performance Metrics:

Elaborate on the significance of using "number of secure clusters" and "load balancing" as performance metrics. Why are these metrics chosen, and how do they relate to the goals of the clustering algorithm?

Figures 5 and 6:

Provide axis labels and legends to enhance the readability of the figures. This will make it easier for readers to interpret the results.

Relation between Transmission Range and Clusters:

Clarify the observed relationship between transmission range and the number of secure clusters. A more detailed explanation would add depth to the interpretation.

Discussion on Simulation Area (Lines 566-572):

Elaborate on the relationship between the simulation area and the number of secure clusters. Why does increasing the simulation area result in more secure clusters?

Numerical Analysis (Lines 575-622):

Provide a bit more context around the numerical analysis, especially the percentages mentioned in Table 7. What do these percentages represent?

Load Balance Factor (LBF) (Lines 602-616):

Equation 11:

Explain the significance of ϕc and how it influences the load balance factor. A more detailed explanation of the formula would be helpful.

Load Balance Results (Fig 10):

Add labels and legends to the figure for clarity. Also, explain the significance of the results and how they contribute to the overall assessment of the algorithm.

Conclusion (Lines 625-646):

Integration of Emerging Technologies:

Provide more details on how the proposed algorithm could adapt to emerging technologies such as 5G, 6G, and Flying Adhoc Networks (FANET). What specific challenges and optimizations are anticipated in these scenarios?

Overall Suggestions:

Consistent Terminology:

Ensure that terminology is used consistently throughout the section. For example, use either "secure clusters" or "optimal clusters" consistently.

Visual Aids:

Consider adding visual aids, such as flowcharts or diagrams, to illustrate the algorithm's steps and the experimental setup.

Interpretation Guidance:

Provide more guidance on interpreting the figures and tables. Explain the observed trends and their implications for the algorithm's performance.

Statistical Analysis:

If applicable, consider including statistical analysis of results (e.g., standard deviations) to quantify the robustness and reliability of the algorithm.

Future Work:

Expand on potential directions for future work, especially in terms of refining the algorithm or addressing specific challenges in vehicular networks.

Reference

I recommend considering the article titled "Optimized traffic flow prediction based on cluster formation and reinforcement learning" as it aligns well with the proposed system.

Reviewer #2: There is no security analysis available in this manuscript. The authors are instructed to analysis the following papers to prepare the security analysis part.

1. Dual authentication and key management techniques for secure data transmission in vehicular ad hoc networks

2. An anonymous batch authentication and key exchange protocols for 6G enabled VANETs

3.CPAV: Computationally efficient privacy preserving anonymous authentication scheme for vehicular ad hoc networks

4.Computationally efficient privacy preserving authentication and key distribution techniques for vehicular ad hoc networks

6. PLOS authors have the option to publish the peer review history of their article (what does this mean?). If published, this will include your full peer review and any attached files.

Reviewer #1: No

Reviewer #2: No

---

## [Author Response · Author response to Decision Letter 0]

14 Feb 2024

Comments from the editors and reviewers:

-Reviewer:1

 Comments:

Comment 1: The article lacks proper organization with the absence of headings and subheadings throughout its content. 

Abstract:

i. Clarify the specific meta-heuristic algorithm employed by AOACNET.

ii. Provide more details on the threshold condition for authenticating nodes during cluster formation.

iii. Specify the nature of the extensive simulation results and the metrics used for evaluation.

Answer:

We are thankful to the respectable reviewer for his concern for improvement in the manuscript. The format, language and organization of the paper is improved as per the direction of the respectable reviewer. 

Moreover, in the revised manuscript, we clarify the use of specific meta-heuristic algorithm for the proposed clustering algorithm. We also provide a detail on threshold value which is set to 0. If a vehicle performance value is greater than the threshold value, the vehicle will consider an authentic vehicle and will be able to be a part of the cluster. In addition, we also specify the evaluation metric and simulation parameters for extensive simulation.

Comment 2: Introduction : 

i. Elaborate on the unique challenges posed by the dynamic network topology in VANETs.

ii. Include a concise comparison with existing clustering methods in VANETs to emphasize the innovation of AOACNET.

iii. The section provides a comprehensive overview of the challenges in Vehicular Adhoc Networks (VANET) and the importance of secure vehicular clustering. However, consider breaking down the lengthy paragraphs into smaller ones for improved readability

Answer: We are thankful to the respectable reviewer for his concern for improvement in the manuscript. The introduction section is revised as per the direction of the respectable reviewer. 

Comment 3: Motivation and Problem Statement:

The motivation for secure vehicular clustering is well-stated, particularly emphasizing the impact of malicious nodes on passenger safety. Consider providing specific examples or scenarios to illustrate the severity of the problem.

Answer:

We are thankful to the respectable reviewer for his positive comments and his concern for improvement in the manuscript. The scenarios are provided as per the direction of respectable reviewer. 

Comment 4: Introduction to Vehicular Clustering:

The introduction to vehicular clustering is informative. However, you may want to elaborate more on the benefits and challenges of clustering in VANETs. How does clustering address the issues of dynamic network topology and fast vehicle movement?

Answer:

We are thankful to the respectable reviewer for his positive comments and his concern for improvement in the manuscript. In the revised manuscript, we discuss the benefits and challenges of vehicular clustering. In addition, we also discuss how clustering address the issues of dynamic network topology.

Comment 5: Importance of Optimal Clustering:

The importance of optimal clustering is highlighted, touching on factors like communication efficiency and resource utilization. Consider providing more concrete examples or studies that demonstrate the impact of optimal clustering on VANET performance.

Answer:

We are thankful to the respectable reviewer for his concern for improvement in the manuscript. In the revised manuscript we provide more details regarding the impact of optimal clustering on VANET performance. 

Comment 6: Introduction to AOA (Arithmetic Optimization Algorithm)

The introduction to AOA is clear, but you might want to briefly explain why AOA is chosen over other meta-heuristic algorithms. What specific characteristics of AOA make it suitable for vehicular clustering?

Answer:

We are thankful to the respectable reviewer for his concern for improvement in the manuscript. In the revised manuscript, we briefly add the characteristics of AOA that make it effective method for vehicular clustering. 

Comment 7: Clarity and Organization: 

The literature review provides a comprehensive overview of clustering protocols in vehicular networks. However, consider breaking down the section into smaller subsections for better organization, making it easier for readers to navigate through the content.

Answer:

We are thankful to the respectable reviewer for his positive comment and his concern for improvement in the manuscript. Basically the literature review is already divided into section and subsection for better organization and clarity. Broadly it is divided into three section i.e.. General purpose clustering, domain-specific clustering algorithm and meta-heuristic algorithm. The domain specific clustering algorithm is further divided into four subsection i.e. Cluster-based Routing algorithm, MAC-based clustering algorithm, secure clustering algorithm and QoS-aware clustering algorithm. Each of these subsection includes various high qualities papers. 

Comment 8: Taxonomy of clustering protocol 

The classification of clustering protocols into general-purpose and domain-specific categories is clear. Consider providing a brief summary or transition statement before introducing Fig 3 to prepare the reader for the taxonomy.

Answer:

We are thankful to the respectable reviewer for his positive comment and his concern for improvement in the manuscript. The taxonomy of clustering protocol shows the division of clustering algorithm into three division i.e., general-based, domain specific and meta-heuristic and their subdivision that are already briefly discuss just before the inclusion of figure 3. However, in the revised manuscript we add a brief statement about meta-heuristic clustering algorithm. 

Comment 9: Fig 3- Classification of clustering protocol 

The taxonomy figure is a valuable addition. However, ensure that the figure is adequately explained in the text. Each category and its significance should be briefly discussed.

Answer:

We are thankful to the respectable reviewer for his positive comment and his concern for improvement in the manuscript. All the categories and sub categories of Fig 3 are significantly explained in the manuscript. 

Comment 10: General Purpose Clustering Protocol: 

The descriptions of general-purpose clustering algorithms are informative. Consider adding a sentence or two after each algorithm to highlight its strengths or limitations in the context of vehicular networks.

Answer:

We are thankful to the respectable reviewer for his positive comment and his concern for improvement in the manuscript. In the revised manuscript we highlight the strengths and weakness of each general purpose clustering protocol as suggested by the respectable reviewer.

Comment 11: Domain specific Clustering Protocol: 

The categorization of domain-specific clustering protocols into routing, security, MAC, and QoS is effective. Ensure that each sub-section provides a concise yet comprehensive overview of the protocols within that domain.

Answer:

We are thankful to the respectable reviewer for his positive comment and his concern for improvement in the manuscript. In the revised manuscript, we provide the concise and comprehensive detail of each sub-category as per the direction of the respectable reviewer.

Comment 12: Routing Protocol 

The description of clustering in the context of vehicular routing is clear. Consider adding a brief sentence about the importance of clustering in improving routing efficiency in VANETs.

Answer:

We are thankful to the respectable reviewer for his positive comment and his concern for improvement in the manuscript. In the revised manuscript, we briefly discuss the importance of clustering in improving routing efficiency in VANET as per the direction of the respectable reviewer.

Comment 13: Security Protocol 

The overview of security protocols is well-structured. However, for each protocol, consider providing a sentence about how clustering enhances security in vehicular networks.

Answer:

We are thankful to the respectable reviewer for his positive comment and his concern for improvement in the manuscript. This section is revised as per the instruction of the respectable reviewer.

Comment 14: Cluster-based MAC Protocol 

The section on MAC protocols is comprehensive. Consider briefly explaining how cluster-based MAC protocols address challenges like fair channel access in vehicular networks.

Answer:

We are thankful to the respectable reviewer for his positive comment and his concern for improvement in the manuscript. In the revised manuscript, we explain the procedure of fair channel access using cluster-based MAC protocol. 

Comment 15: QoS assurance Protocol 

The discussion on QoS assurance protocols is well-done. Consider emphasizing the role of clustering in maintaining QoS metrics and how it contributes to the overall network performance.

Answer:

We are thankful to the respectable reviewer for his positive comment and his concern for improvement in the manuscript. In the revised manuscript, we explain the role of clustering in maintaining the QoS of vehicular network as per the instruction of the respectable reviewer.

Comment 16: Meta-heuristic based clustering algorithm

The explanation of meta-heuristic algorithms is clear. Consider adding a sentence after each algorithm to highlight its specific advantages in the context of clustering for vehicular networks.

Answer:

We are thankful to the respectable reviewer for his positive comment and his concern for improvement in the manuscript. In the revised manuscript, we highlight the specific advantage of each meta-heuristic based clustering algorithm as per direction of the respectable reviewer.

Comment 17: Arithmetic Optimization Algorithm 

The introduction of AOA is well-described. Consider adding a sentence or two about why AOA is chosen as the focus of the paper and how it differs from other meta-heuristic algorithms.

Answer:

We are thankful to the respectable reviewer for his positive comment and his concern for improvement in the manuscript. In the introduction section, we discuss the same points in details that briefly explain the reasons behind selection of AOA over other meta-heuristic algorithms. In this section, we only describe the working of AOA algorithm. 

Comment 18: Distribution behavior of AOA 

The explanation of the distribution behavior of AOA is clear. Consider providing a brief example or analogy to help readers conceptualize the diversification and intensification phases.

Answer:

We are thankful to the respectable reviewer for his positive comment and his concern for improvement in the manuscript. In the revised manuscript, we explain the role of clustering in maintaining the QoS of vehicular network as per the instruction of the respectable reviewer. 

Comment 19: Explanation of MOA Function:

The MOA function is introduced, but there's no detailed explanation of its purpose or significance. Provide more context on how it contributes to the search space creation.

Answer:

We are thankful to the respectable reviewer for his concern for improvement in the manuscript. Math Optimizer function is a linear function whose value decides the usage of two important features of AOA algorithm i.e., exploration and exploitation phase. The value of the MOA is updated using eq 1 of paper in each iteration from 0.2 to 0.9. The purpose of the MOA is to give an equal chance to the algorithm to choose between exploration or exploitation search phase. In case of rand > MOA, the algorithm uses exploration phase to explore the search space for diverse potentials solutions and in other case, algorithm employs exploitation phase to conduct the search for optimal solution within the search space. 

Comment 20: Explanation of Search Space

Clarify how the search space is represented and how candidate solutions move within this space. A visual representation or example would be beneficial. 

Answer:

We are thankful to the respectable reviewer for his kind suggestion and his concern for improvement in the manuscript. The search space is created with a set of candidate solution. Each point in the search space represent a candidate solution which is encapsulated with the cluster matrix. Each candidate solution has a different set of vehicular clusters which is evaluated using the fitness function. In each iteration, the objective function determines the optimal solution. Afterward the candidate solution moves around the best optimal solution using exploration or exploitation search phase. 

Comment 21: Explanation of secure cluster formation (Lines 416-443):

The section is detailed, but consider breaking down complex processes, such as Roulette Wheel Selection (RWS), into smaller steps for better understanding.

Answer:

We are thankful to the respectable reviewer for his concern for improvement in the manuscript. The subsection is revised as per the direction of the respectable reviewer. 

Comment 22: Roulette Wheel Selection 

Provide a bit more context or a simple example of how RWS works in the context of secure cluster formation.

Answer:

We are thankful to the respectable reviewer for his concern for improvement in the manuscript. We provide more details on RWS in the context of secure cluster formation. 

Comment 23: Explanation of objective function 

Offer a brief explanation of why a multi-objective fitness function is chosen and how ψ and μ contribute to the evaluation of solutions. 

Answer:

We are thankful to the respectable reviewer for his concern for improvement in the manuscript. The multi-objective function is chosen, the vehicular clustering is considered as multi-objective problem, where multiple objectives need to be maximized or minimized simultaneously. In this problem, we use two objectives, one is distance neighbor and the other is delta difference. Both objectives are necessary for vehicular cluster optimization, and for communication efficiency.

Comment 24: Delta Difference (Equation 7)

Provide a bit more explanation or intuition behind the delta difference concept.

Answer:

We are thankful to the respectable reviewer for his concern for improvement in the manuscript. In the revised manuscript, we provide the details of delta difference. Basically, the value for the delta varies subject to the network condition i.e., for a highly dense network, the delta value will be high and vice versa. The minimum value of delta difference is required to achieve optimization, scalability, and reliability of the communication protocol. Each cluster tries to cover maximum number of cluster members but below delta value in an attempt to optimize the number of clusters. 

Comment 25: Clarification on Exploration and Exploitation:

Clearly explain the roles of exploration and exploitation in the context of repositioning. It might help to provide an example or visual aid.

Answer:

We are thankful to the respectable reviewer for his concern for improvement in the manuscript. Exploration and exploitation phases are the two key features of meta-heuristic algorithms. Exploration, randomly explore the regions of the search space for wide range of potential solutions. In the exploration search phase, the algorithm randomly move within the search space to discover the regions that may contain potential solution. Exploitation, intensify the regions found by the exploration search phase. In the exploitation phase, the algorithm move to regions found in the previous phase, to fine-tune the solution. 

Comment 26: Mathematical Notation

Ensure that all symbols used in equations are defined or referenced in Table 1 for clarity.

Answer:

We are thankful to the respectable reviewer for his concern for improvement in the manuscript. All the symbols are referenced and defined clearly. 

Comment 27: Explanation of Exploration and Exploitation:

Offer more insight into why certain operators are considered explorative or exploitative and how they affect the search process.

Answer:

We are thankful to the respectable reviewer for his concern for improvement in the manuscript. In arithmetic operation, certain operators like multiplication and division introduces large changes to the variable’s values. These operators have high dispersion and distributed 

---

## [Decision Letter · Decision Letter 1]

2 Apr 2024

PONE-D-23-43275R1Arithmetic Optimization based Secure Intelligent Clustering Algorithm for Vehicular Adhoc NetworkPLOS ONE

Dear Dr. Ali,

Thank you for submitting your manuscript to PLOS ONE. After careful consideration, we feel that it has merit but does not fully meet PLOS ONE’s publication criteria as it currently stands. Therefore, we invite you to submit a revised version of the manuscript that addresses the points raised during the review process.

We look forward to receiving your revised manuscript.

Kind regards,

Yuanguo Bi, PhD

Academic Editor

PLOS ONE

**Additional Editor Comments:**

Please revise the manuscript according to the comments from reviewers.

Reviewers' comments:

Reviewer's Responses to Questions

**Comments to the Author**

Reviewer #1: All comments have been addressed

Reviewer #2: (No Response)

2. Is the manuscript technically sound, and do the data support the conclusions?

Reviewer #1: Yes

Reviewer #2: No

3. Has the statistical analysis been performed appropriately and rigorously? 

Reviewer #1: Yes

Reviewer #2: No

4. Have the authors made all data underlying the findings in their manuscript fully available?

Reviewer #1: Yes

Reviewer #2: (No Response)

5. Is the manuscript presented in an intelligible fashion and written in standard English?

Reviewer #1: Yes

Reviewer #2: (No Response)

6. Review Comments to the Author

Reviewer #1: The author has diligently taken into consideration and appropriately addressed all the comments therefore, I recommend accepting the revised manuscript.

Reviewer #2: No proper security analysis with respect to attack scenarios. Refer the following papers for security analysis.

1.EAAP: Efficient anonymous authentication with conditional privacy-preserving scheme for vehicular ad hoc networks

2.Dual authentication and key management techniques for secure data transmission in vehicular ad hoc networks

3.An efficient anonymous mutual authentication technique for providing secure communication in mobile cloud computing for smart city applications

7. PLOS authors have the option to publish the peer review history of their article (what does this mean?). If published, this will include your full peer review and any attached files.

Reviewer #1: No

Reviewer #2: No

---

## [Author Response · Author response to Decision Letter 1]

8 Apr 2024

-Reviewer:1

Comment 1: The author has diligently taken into consideration and appropriately addressed all the comments therefore, I recommend accepting the revised manuscript.

Answer:

We are thankful to the respectable reviewer for considering the comments and accepting the revised manuscript. 

-Reviewer:2

 Comments:

Comment 1: No proper security analysis with respect to attack scenarios. Refer the following papers for security analysis.

Answer:

We are thankful to the respectable reviewer for his concern for improvement in the manuscript. We carefully consider the comments of the respectable reviewer and analyze the security of the proposed algorithm concerning different attack scenarios i.e., resistance against impersonation attack, replay attack, Sybil node attack, and bogus message attack. The dynamic performance attribute of each vehicle and the valid timestamp and cryptographic nonce of each message enables the proposed algorithm to effectively resist security attacks.

Moreover, we also analyze the performance of the proposed algorithm in terms of computational time which refers to the time taken by the algorithm to authenticate a single vehicle. The simulation results declare that with minimal calculations, the proposed algorithm consumes less time as compared to other schemes.

---

## [Decision Letter · Decision Letter 2]

13 Aug 2024

PONE-D-23-43275R2Arithmetic Optimization based Secure Intelligent Clustering Algorithm for Vehicular Adhoc NetworkPLOS ONE

Dear Dr. Ali,

Thank you for submitting your manuscript to PLOS ONE. After careful consideration, we feel that it has merit but does not fully meet PLOS ONE’s publication criteria as it currently stands. Therefore, we invite you to submit a revised version of the manuscript that addresses the points raised during the review process.

We look forward to receiving your revised manuscript.

Kind regards,

Zhiquan Liu, Ph.D.

Academic Editor

PLOS ONE

Journal Requirements:

**Additional Editor Comments:**

Almost all reviewers are very positive about the contribution of this paper, and also point out some constructive comments. Please revise it according to the opinions as soon as possible, and submit the revised version to ensure that this paper can be published quickly.

Reviewers' comments:

Reviewer's Responses to Questions

**Comments to the Author**

1. If the authors have adequately addressed your comments raised in a previous round of review and you feel that this manuscript is now acceptable for publication, you may indicate that here to bypass the “Comments to the Author” section, enter your conflict of interest statement in the “Confidential to Editor” section, and submit your "Accept" recommendation.

Reviewer #1: All comments have been addressed

Reviewer #2: All comments have been addressed

Reviewer #3: All comments have been addressed

Reviewer #4: (No Response)

2. Is the manuscript technically sound, and do the data support the conclusions?

Reviewer #1: Yes

Reviewer #2: Partly

Reviewer #3: Yes

Reviewer #4: Yes

3. Has the statistical analysis been performed appropriately and rigorously? 

Reviewer #1: Yes

Reviewer #2: Yes

Reviewer #3: Yes

Reviewer #4: Yes

4. Have the authors made all data underlying the findings in their manuscript fully available?

Reviewer #1: Yes

Reviewer #2: Yes

Reviewer #3: Yes

Reviewer #4: Yes

5. Is the manuscript presented in an intelligible fashion and written in standard English?

Reviewer #1: Yes

Reviewer #2: Yes

Reviewer #3: Yes

Reviewer #4: Yes

6. Review Comments to the Author

**Reviewer #1:** Please cite the following articles that are related to the proposed article: 1. Blockchain-based batch authentication protocol for Internet of Vehicles 2. Blockchain-based mutual-healing group key distribution scheme in unmanned aerial vehicles ad-hoc network.

**Reviewer #2:** No comments from my side. This paper can be accepted in its current form...No comments from my side. This paper can be accepted in its current form...

**Reviewer #3: **1. In-depth Discussion on Security Analysis: "Please further discuss the performance of the proposed algorithm under various malicious attacks in the security analysis section. For example, can the algorithm effectively respond to rapid changes in malicious nodes in highly dynamic environments? Additionally, considering the potential diversity of malicious node attack patterns, could you add an analysis of the algorithm's ability to detect other types of attacks, such as node hijacking?"

2. Algorithm Efficiency Comparison: "In the comparison of computation time, it is recommended to include some of the latest commonly used security algorithms in addition to the existing comparison algorithms. This will help to more comprehensively demonstrate the efficiency advantages of the proposed algorithm."

3. Real-world Scenario Testing for Latency and Network Overhead: "In the section on end-to-end latency and network overhead, it is suggested to add some test data from real network environments. These data can validate the performance of the proposed algorithm in real-world applications, especially in networks of different scales and densities."

4. Expansion of Future Work: "In the conclusion section, the directions for future work can be described more specifically, such as how to optimize the algorithm by integrating machine learning techniques, or how to further improve the algorithm to meet the specific needs of 5G and 6G networks. Detailed implementation plans and expected outcomes would help to enhance the feasibility of future work."

5. Explanation of Figures and Data: "The results presented in Figures 11, 12, and 13 should include more detailed explanations, such as the specific parameter settings of different algorithms and the network environment conditions. This information will help readers better understand the data and comparison results shown in the figures."

6.Writing and Formatting:"The overall writing of the article is smooth, but there are some minor errors, such as 'VAENT' should be 'VANET' and 'has' should be 'have.' It is recommended to carefully proofread the text. The format of the references needs to be consistent, and please ensure that all cited references are correctly listed."

**Reviewer #4:** In this work, the authors explore vehicular clustering as combinatorial optimization problem using the Arithmetic Optimization Algorithm (AOA) to address the issue of the 882 883 884 885 scalability and reliability of the vehicular network. This is a meaningful work. Following are some suggestions to the authors to improve the quality of the paper. The chapter numbering of the paper needs to be standardized. More details of the experimental environment should be explained. The abstract is too long. The author needs to add the latest literatures and analyze them, such as ppru: a privacy-preserving reputation updating scheme for cloud-assisted vehicular networks, trove: a context awareness trust model for vanets using reinforcement learning. The captions of Figure5 to Figure13 should be rewritten. The conclusion is too long, please refine it. Please give specific strengths of the paper in the abstract and conclusion, expressed by specific numerical indicators.

7. PLOS authors have the option to publish the peer review history of their article (what does this mean?). If published, this will include your full peer review and any attached files.

Reviewer #1: **Yes: **RAJKUMAR S C

Reviewer #2: No

Reviewer #3: **Yes: **Kai Yang

Reviewer #4: No

---

## [Author Response · Author response to Decision Letter 2]

19 Aug 2024

Authors’ response to the reviewers’ comments

Title: Arithmetic Optimization based Secure Intelligent Clustering Algorithm for Vehicular Ad-hoc Network

Journal: PLOS ONE

The authors are thankful to the reviewers and the associate editor for providing their useful and constructive comments. We have taken carefully your valuable and prolific suggestions into consideration while preparing our revision. The modified paper got much improvement and clarity by incorporating the comments. We hope that the enclosed revised paper will meet the requirements suggested by the reviewers and the associate editor. 

Note: There are two versions of the manuscript i.e. manuscript- (clean copy) and manuscript- (track changes). 

Comments from the editors and reviewers:

-Reviewer:1

Comment 1:Please cite the following articles that are related to the proposed article: 1. Blockchain-based batch authentication protocol for Internet of Vehicles 2. Blockchain-based mutual-healing group key distribution scheme in unmanned aerial vehicles ad-hoc network.

Answer:

We are thankful to the respectable reviewer for his concern for improvement in the manuscript. The suggested articles are cited in the Literature review section (subsection –secure clustering algorithm) as per the suggestion of the respectable reviewer.

-Reviewer:2

Comment 1: No comments from my side. This paper can be accepted in its current form.

Answer:

We are thankful to the respectable reviewer for accepting the revised manuscript.

Reviewer:3

Comment 1: In-depth Discussion on Security Analysis: "Please further discuss the performance of the proposed algorithm under various malicious attacks in the security analysis section. For example, can the algorithm effectively respond to rapid changes in malicious nodes in highly dynamic environments? Additionally, considering the potential diversity of malicious node attack patterns, could you add an analysis of the algorithm's ability to detect other types of attacks, such as node hijacking?"

Answer: We are thankful to the respectable reviewer for his concern for improvement in the manuscript. 

The proposed algorithm is designed to operate efficiently in a highly dynamic vehicular network. The performance attribute of each vehicle is dynamically updated based on real-time communication, and even after the selection of a cluster member, the RSU continuously monitors the behavior of cluster member. This includes, observing irregular patterns, sudden message spikes and sudden variation is performance attribute). Therefore due to real-time continuous monitoring and dynamic update mechanism of performance value, the proposed algorithm is able to respond to rapid changes in malicious nodes. 

The proposed algorithm is versatile in detecting a range of malicious attacks including node hijacking attack. The security analysis of node hijacking attack is included in the manuscript as per the suggestion of the respectable reviewer. 

Comment 2: Algorithm Efficiency Comparison: "In the comparison of computation time, it is recommended to include some of the latest commonly used security algorithms in addition to the existing comparison algorithms. This will help to more comprehensively demonstrate the efficiency advantages of the proposed algorithm."

Answer: We are thankful to the respectable reviewer for his kind suggestion. The set of benchmark algorithms in terms of computational time is increased as per the suggestion of the respectable reviewer.

Comment 3: Real-world Scenario Testing for Latency and Network Overhead: "In the section on end-to-end latency and network overhead, it is suggested to add some test data from real network environments. These data can validate the performance of the proposed algorithm in real-world applications, especially in networks of different scales and densities."

Answer: We are thankful to the respectable reviewer for his kind suggestion. The proposed algorithm is simulated in a specialized MATLAB tool version 2021a through various realistic network scenarios with varying network density, transmission range and network area. We design multiple realistic scenarios with different network size (small, medium and large) i.e., 30-60 nodes in a network area (1-2 km) to validate the performance of the proposed algorithm. Various performance metrics such as latency and network overhead are recorded by varying transmission range in a simulation area of 1x1 km. The result demonstrate the efficacy of the proposed algorithms in terms of designated performance metrics.

Comment 4: Expansion of Future Work: "In the conclusion section, the directions for future work can be described more specifically, such as how to optimize the algorithm by integrating machine learning techniques, or how to further improve the algorithm to meet the specific needs of 5G and 6G networks. Detailed implementation plans and expected outcomes would help to enhance the feasibility of future work."

Answer: We are thankful to the respectable reviewer for his kind concern for improvement in the manuscript. The conclusion section is revised as per the direction of the respectable reviewer.

Comment 5: Explanation of Figures and Data: "The results presented in Figures 11, 12, and 13 should include more detailed explanations, such as the specific parameter settings of different algorithms and the network environment conditions. This information will help readers better understand the data and comparison results shown in the figures."

Answer: We are thankful to the respectable reviewer for his kind concern for improvement in the manuscript. More details regarding parameter settings and network environment conditions are included as per the direction of the respectable reviewer. Table 6 (simulation parameters) is also updated accordingly.

Comment 6: Writing and Formatting: “The overall writing of the article is smooth, but there are some minor errors, such as 'VAENT' should be 'VANET' and 'has' should be 'have.' It is recommended to carefully proofread the text. The format of the references needs to be consistent, and please ensure that all cited references are correctly listed."

Answer: We are thankful to the respectable reviewer for his kind concern for improvement in the manuscript. All the typo errors along with format of the references are corrected as per the recommendation of the respectable reviewer.

Reviewer 4:

Comment 1: In this work, the authors explore vehicular clustering as combinatorial optimization problem using the Arithmetic Optimization Algorithm (AOA) to address the issue of the 882 883 884 885 scalability and reliability of the vehicular network. This is a meaningful work. Following are some suggestions to the authors to improve the quality of the paper. The chapter numbering of the paper needs to be standardized. More details of the experimental environment should be explained. The abstract is too long. The author needs to add the latest literatures and analyze them, such as ppru: a privacy-preserving reputation updating scheme for cloud-assisted vehicular networks, trove: a context awareness trust model for VANETs using reinforcement learning. The captions of Figure5 to Figure13 should be rewritten. The conclusion is too long, please refine it. Please give specific strengths of the paper in the abstract and conclusion, expressed by specific numerical indicators.

Answer: We are thankful to the respectable reviewer for his kind concern for improvement in the manuscript. All the suggestions are taken accordingly and revised the manuscript as per the suggestion of the respectable reviewer. The abstract and conclusion is revised accordingly. Moreover, the references list is also corrected.

Versus

---

## [Editor Report · Decision Letter 3]

21 Aug 2024

Arithmetic Optimization based Secure Intelligent Clustering Algorithm for Vehicular Adhoc Network

PONE-D-23-43275R3

Dear Dr. Ali,

We’re pleased to inform you that your manuscript has been judged scientifically suitable for publication and will be formally accepted for publication once it meets all outstanding technical requirements.

Kind regards,

Prof. Zhiquan Liu

Academic Editor

Jinan University

zqliu@vip.qq.com

https://www.zqliu.com

Additional Editor Comments (optional):

Accept
---

## [Editor Report · Acceptance letter]

2 Sep 2024

PONE-D-23-43275R3 

PLOS ONE

Dear Dr. Ali, 

I'm pleased to inform you that your manuscript has been deemed suitable for publication in PLOS ONE. Congratulations! Your manuscript is now being handed over to our production team.

Kind regards, 

on behalf of

Professor Zhiquan Liu 

Academic Editor

PLOS ONE